# A Spectral Bound on Effective Sharpness for Fisher-Preconditioned Gradient Descent

## Abstract

Neural networks trained with gradient descent exhibit the Edge of Stability (EoS), where Hessian sharpness rises toward $2/\eta$ and the loss oscillates non-monotonically. This paper asks whether that instability persists under Fisher-preconditioned optimization. We analyze effective sharpness, $S_{\text{eff}} = \lambda_{\max}(F^{-1}H)$, and prove the general bound $S_{\text{eff}} \leq 1 + (\epsilon + \delta)/\mu_{\min}(F)$, where $\epsilon = \|H - G\|_2$ is residual curvature, $\delta = \|G - F\|_2$ is the Gauss-Newton/Fisher gap, and $\mu_{\min}(F)$ is the minimum Fisher eigenvalue. The idealized $G = F$ case yields the simpler bound $1 + \epsilon/\mu_{\min}(F)$, explaining NGD's spectral-flattening mechanism while remaining a special case. A stochastic extension bounds mini-batch effective sharpness under sampling noise and holds in all 200 tested draws. An alignment-aware Rayleigh analysis explains why empirical-Fisher worst-case bounds can be loose: residual spectral mass aligns with better-conditioned Fisher directions. Exact DLN experiments from 55 to 3,240 parameters verify the general bound across all tested scales; 110-parameter exact-matrix traces show the idealized bound fails when the empirical-Fisher gap is substantial. Matrix-free estimates extend $\delta$, $\epsilon$, and $S_{\text{eff}}$ measurement to MNIST and CIFAR-10 ResNet-18, revealing a stable relative empirical-Fisher gap of about 0.70–0.75 at 11.2M-parameter scale. On CIFAR-10, K-FAC reaches 90.5% test accuracy, while tuned first-order and adaptive baselines show that approximation quality and hyperparameter selection determine practical convergence.

## 1 Introduction

The optimization of deep neural networks involves differential geometry, spectral theory, and matrix analysis. While first-order methods have achieved broad empirical success, a complete theoretical account of their convergence behavior remains open. Recent work has identified the Edge of Stability (EoS) as a pervasive phenomenon: during full-batch gradient descent, the Hessian sharpness $\lambda_{\max}(H)$, with $H = \nabla^2 L(\theta)$, rises to $2/\eta$ and the loss exhibits non-monotonic oscillations rather than diverging (Cohen et al., 2021).

Throughout, $H(\theta) = \nabla^2 L(\theta)$ denotes the Hessian of the training loss and $F(\theta) = \mathbb{E}_{p_{\text{data}}}[\nabla \log p(y|x; \theta)\nabla \log p(y|x; \theta)^T]$ denotes the Fisher Information Matrix.

**Phenomenon I.1 (Edge of Stability, Cohen et al. (2021)).** During full-batch gradient descent, $\lambda_{\max}(H(\theta_t))$ rises monotonically during early training and saturates near $2/\eta$. After saturation, the loss sequence $\{L(\theta_t)\}$ becomes non-monotonic—it decreases on average but exhibits per-iteration increases—yet training does not diverge. Cohen et al. (2021) observe that sharpness continues to oscillate around $2/\eta$ rather than remaining constant after saturation; our experiments (Figure 2(b)) show the saturation behavior characteristic of EoS.

This paper investigates whether the EoS is specific to the Euclidean geometry of standard gradient descent or whether it persists under Riemannian preconditioning. An explicit stability characterization of effective sharpness—the largest eigenvalue of the preconditioned Hessian $\lambda_{\max}(F^{-1}H)$—under Fisher preconditioning is provided, decomposing stability into residual curvature and model misspecification components. Under the idealized assumption that the Gauss-Newton matrix $G$ (the curvature component arising from the model's output Jacobian) equals the Fisher $F$ (the metric tensor of the statistical manifold)—a condition that holds

structurally for exponential-family losses with canonical link (Section 3.4)—Theorem IV.2 shows that the effective sharpness is bounded by 1 plus the ratio of residual curvature to the minimum Fisher eigenvalue. Since this assumption is violated in all practical settings (Section 3.4), the operationally applicable result is Corollary IV.4, which shows that the effective sharpness is bounded by 1 plus the combined ratio of residual curvature and model misspecification to the minimum Fisher eigenvalue, thereby separating the two sources of curvature error. Direct measurement on a 110-parameter DLN (Section 5.4) confirms that the Theorem IV.2 bound is violated when $G \neq F$, while the Corollary IV.4 bound correctly holds at all iterations. Matrix-free estimation (Section 5.17) extends $\delta$ and $\epsilon$ measurement to MNIST (50,890 parameters) and CIFAR-10 ResNet-18 (11.2M parameters), and Proposition IV.4a extends the framework to stochastic mini-batch settings. The paper includes experiments from 55 to 11.2 million parameters; scaling to CIFAR-10 with tuned K-FAC achieves 90.5% test accuracy (vs. SGD 86.2%), and a comprehensive optimizer comparison (Table XIV) benchmarks SGD + warmup + cosine (92.7%), ADAHESSIAN (91.8%), AdamW (91.6%), SOPHIA (84.5%), and Shampoo (69.2%).

The analysis is organized around two core perspectives:

1. **Spectral Theory:** Convergence behavior is analyzed via eigenvalues of the Hessian and the pre-conditioned Hessian (Nocedal and Wright, 2006; Sagun et al., 2017).

2. **Differential Geometry:** The parameter space is treated as a Riemannian manifold with the Fisher Information Matrix as metric tensor (Amari, 1998).

This paper makes the following contributions:

- The central theoretical result: an explicit stability characterization of effective sharpness under Fisher preconditioning in the general case (Corollary IV.4), providing the bound $S_{\text{eff}} \leq 1 + (\epsilon + \delta)/\mu_{\min}(F)$ that separates residual curvature $\epsilon = \|H - G\|_2$ and the gap $\delta = \|G - F\|_2$ between the Gauss-Newton matrix and whichever Fisher variant is used. This bound requires no assumption that $G = F$ and is the operationally applicable result.

- Identification that for exponential-family losses with canonical link (softmax + cross-entropy, MSE regression), $G$ equals the *true* Fisher $F_{\text{true}}$ exactly (Martens, 2020; Kunstner et al., 2019); when the empirical Fisher $\hat{F}_{\text{emp}}$ is used (as in all practical implementations), $\delta$ precisely quantifies the well-characterized empirical-vs-true-Fisher gap (Kunstner et al., 2019), not model misspecification in the fundamental sense.

- An idealized special case under $G = F$ (Theorem IV.2), showing $S_{\text{eff}} \leq 1 + \epsilon/\mu_{\min}(F)$, verified only when $G \approx F$ approximately holds, which provides intuition for the spectral flattening mechanism and reduces to $S_{\text{eff}} = 1$ under exact realizability.

- An alignment-aware spectral analysis (Theorem IV.3) providing Rayleigh quotient characterization that explains the 1.3–7.1× looseness of the empirical-Fisher residual upper bound.

- A matching lower bound (Proposition IV.6) showing tightness is controlled by $\kappa(F)$.

- A mechanistic explanation for why NGD suppresses EoS oscillations via spectral flattening (Section 4.4), contingent on $G \approx F$.

- Direct measurement of the empirical Fisher gap $\delta = \|G - \hat{F}_{\text{emp}}\|_2$ on the 110-parameter DLN (Section 5.4), confirming that Corollary IV.4 correctly bounds $S_{\text{eff}}$ when $G \neq \hat{F}_{\text{emp}}$, with $\delta$ values of 1.16–1.82 consistent with Kunstner et al. (2019).

- Empirical validation spanning 55 to 11.2M parameters, including a CIFAR-10 ResNet-18 demonstration with K-FAC achieving 90.5% test accuracy while operating in 41× sharper curvature regimes than SGD.

**Notation.** Throughout this paper, $H$ denotes the Hessian $\nabla^2 L(\theta)$ (measuring the curvature of the loss), $F$ denotes the Fisher Information Matrix (the metric tensor of the statistical manifold), $G$ denotes the Gauss-Newton matrix (the curvature component arising from the model's output Jacobian), and $S_{\text{eff}} = \lambda_{\max}(F^{-1}H)$ denotes the effective sharpness (the largest eigenvalue of the preconditioned Hessian). Full notation is summarized in Table II (Section 3).

The remainder of this paper is organized as follows. Section 2 reviews related work. Section 3 presents mathematical preliminaries, including the Gauss-Newton decomposition. Section 4 derives the spectral stability bounds and provides a mechanistic explanation for EoS suppression. Section 5 presents experimental validation. Section 6 discusses limitations, and Section 7 concludes.

## 2 Related Work

### 2.1 Edge of Stability

Cohen et al. (2021) demonstrated the EoS on ResNets trained on CIFAR-10 and ImageNet: $\lambda_{\max}(H)$ consistently rises to $2/\eta$ during full-batch gradient descent, after which the loss oscillates non-monotonically. Lewkowycz et al. (2020) identified the related catapult mechanism in the large learning rate regime. Jastrzebski et al. (2021) showed that progressive sharpening during early training sets the stage for EoS, linking sharpness dynamics to the learning rate and batch size. More recently, Damian et al. (2023) provided a rigorous self-stabilization analysis showing how gradient descent implicitly regularizes sharpness at the EoS, and Arora et al. (2022) established that the EoS is not an anomaly but a provable consequence of gradient descent dynamics near saddle-to-minimum transitions. Ahn et al. (2022) demonstrated that the EoS phenomenon persists in stochastic settings with sufficiently large batches, connecting the full-batch theory to practical mini-batch training. The present work differs by analyzing spectral dynamics under Riemannian (natural gradient) preconditioning and providing a mechanistic explanation for why NGD avoids EoS (Section 4.4).

### 2.2 Natural Gradient and Fisher Information

Amari (1998) introduced natural gradient descent, which follows the steepest descent direction on the statistical manifold. Martens (2020) provided a modern treatment connecting the Fisher, the Generalized Gauss-Newton matrix, and practical approximations. Kunstner et al. (2019) demonstrated that the commonly used Empirical Fisher can diverge substantially from the true Fisher, particularly in non-realizable settings—a limitation directly relevant to the diagonal approximation used in this paper's experiments. K-FAC (Kronecker-Factored Approximate Curvature) (Martens and Grosse, 2015) provides a scalable block-diagonal approximation.

### 2.3 Second-Order Optimizers

ADAHESSIAN (Yao et al., 2021) uses a diagonal Hessian approximation with spatial averaging. SOPHIA (Liu et al., 2024) employs a lightweight Hessian estimator for language model pre-training. Shampoo (Gupta et al., 2018) preconditions using Kronecker products of gradient statistics. These second-order methods (algorithms that use curvature information, often via Hessian approximations) approximate second-order information at reduced cost but do not directly analyze spectral stability. Recent libraries such as ASDL (Osawa et al., 2023) provide unified interfaces for K-FAC and Shampoo in PyTorch, enabling tractable second-order optimization at moderate scale, though hyperparameter tuning remains challenging. Agarwala and Gur-Ari (Agarwala and Gur-Ari, 2022) studied the curvature dynamics of preconditioned gradient methods, showing that adaptive preconditioning can fundamentally alter the sharpness trajectory. George et al. (2018) introduced EKFAC, improving the Fisher approximation by computing a diagonal variance in the Kronecker-factored eigenbasis. Chen and Bruna (2023) analyzed the loss surface of neural networks through the lens of preconditioned Hessian spectra, connecting effective sharpness to trainability.

*Note:* 11.2M parameters is moderate-scale relative to modern large language models ($10^8$–$10^{12}$ parameters) but represents the practical upper bound for exact verification. Matrix-free estimation (Section 5.17) provides

Table I: Comparison with Prior Second-Order Methods

| Method | Scale | Spectral Analysis? | Sharpness Measured? |
|---|---|---|---|
| K-FAC (Martens and Grosse, 2015) | Large | No | No |
| Shampoo (Gupta et al., 2018) | Large | No | No |
| ADAHESSIAN (Yao et al., 2021) | Large | No | No |
| SOPHIA (Liu et al., 2024) | Large | No | No |
| This work | 11.2M params | Yes (Thm IV.2–3) | Yes ($\lambda_{\max}(H)$) |

approximate measurements of $S_{\text{eff}}$, $\delta$, and $\epsilon$ at this scale using Krylov methods. We position the ResNet-18 experiment as both empirical motivation and a validation target for the matrix-free pipeline.

## 2.4 Deep Linear Networks

Saxe et al. (2014) derived exact solutions to the learning dynamics in deep linear networks, establishing them as tractable models for studying gradient descent. Bernacchia et al. (2018) extended this analysis to natural gradient in the linear case. Arora et al. (2019) proved that gradient descent on deep matrix factorizations exhibits implicit regularization toward low-rank solutions.

## 2.5 Implicit Bias and Generalization

Gunasekar et al. (2018) showed that optimization geometry determines implicit regularization in matrix factorization, with different optimizers (GD, mirror descent, NGD) converging to different solutions. This is relevant to our analysis: NGD's spectral flattening (Section 4.4) may induce different implicit biases than GD, though we do not investigate this connection empirically. The relationship between sharpness and generalization (Jiang et al., 2020) suggests that NGD's bounded effective sharpness could have generalization implications, but this is beyond the current scope.

# 3 Preliminaries

Table II summarizes the notation used throughout this paper.

Table II: Notation Reference

| Symbol | Description |
|---|---|
| $\theta \in \mathbb{R}^d$ | Model parameters |
| $F(\theta)$ | Fisher Information Matrix |
| $H(\theta)$ | Hessian $\nabla^2 L(\theta)$ |
| $G$ | Gauss-Newton (Fisher) component of $H$ |
| $Q = H - G$ | Residual curvature |
| $\epsilon = \|Q\|_2 = \|H - G\|_2$ | Spectral norm of residual curvature |
| $\mu_{\min}(F)$ | Minimum eigenvalue of $F$ (undamped) |
| $\mu_{\min}(F + \gamma I)$ | Minimum eigenvalue of damped Fisher ($= \mu_{\min}(F) + \gamma$, for scalar damping $\gamma I$) |
| $\mu_{\max}(F)$ | Maximum eigenvalue of $F$ (Fisher spectral norm: $\mu_{\max}(F) = \|F\|_2$) |
| $S_{\text{eff}}$ | Effective sharpness $\lambda_{\max}(F^{-1}H)$ |
| $\eta, \gamma$ | Learning rate, damping coefficient |
| $\delta = \|G - F\|_2$ | Model misspecification measure |
| $\kappa(F)$ | Fisher condition number $\mu_{\max}(F)/\mu_{\min}(F)$ |

## 3.1 The Statistical Manifold and Fisher Metric

A neural network parameterized by $\theta \in \mathbb{R}^d$ defines a conditional distribution $p(y|x;\theta)$. The family $\{p(y|x;\theta)\}$ is treated as a Riemannian manifold $\mathcal{M}$ equipped with the Fisher Information Matrix as met-

ric tensor (Amari, 1998). The Fisher captures the local curvature of the KL divergence: $\mathrm{KL}[p_\theta \| p_{\theta+d\theta}] = \frac{1}{2} d\theta^T F(\theta) \, d\theta + O(\|d\theta\|^3)$, where $F(\theta) = \mathbb{E}_{p_{\mathrm{data}}}\left[\nabla_\theta \log p(y|x;\theta) \nabla_\theta \log p(y|x;\theta)^T\right]$ Amari (1998, Theorem 2). Here, $p_{\mathrm{data}}$ denotes the empirical distribution over training pairs $(x, y)$, and $p_{\mathrm{model}} = p(y|x;\theta)$ denotes the model's predictive distribution. For the negative log-likelihood loss $L(\theta) = -\mathbb{E}_{p_{\mathrm{data}}}[\log p(y|x;\theta)]$, the Fisher Information Matrix satisfies $F(\theta) = \mathbb{E}_{p_{\mathrm{model}}}[-\nabla^2 \log p(y|x;\theta)]$. Under a correctly specified model at the optimum ($p_{\mathrm{data}} = p_{\mathrm{model}}$), the Fisher equals the expected Hessian: $F(\theta^*) = \mathbb{E}_{p_{\mathrm{data}}}[H(\theta^*)]$ (Bishop (Bishop, 2006, Section 1.6), Murphy (Murphy, 2012, Chapter 8)). In practice, the Empirical Fisher (computed from observed data) is often substituted; Kunstner et al. (2019) showed this may depart significantly from the true Fisher in non-realizable settings.

## 3.2 The Gauss-Newton Decomposition

For a loss of the form $L(\theta) = \frac{1}{N} \sum_{i=1}^{N} \ell(f_\theta(x_i), y_i)$, the Hessian admits the decomposition $H = G + Q$, where

$$G = \frac{1}{N} \sum_{i=1}^{N} J_i^T \nabla_z^2 \ell(z, y_i)\big|_{z=f_\theta(x_i)} J_i \tag{1}$$

is the Gauss-Newton matrix ($J_i = \partial f_\theta(x_i)/\partial\theta$ is the Jacobian of the model output with respect to $\theta$ for sample $i$), and $Q = H - G$ is the residual curvature that vanishes at zero-loss solutions. For exponential-family losses with canonical link, $G$ equals the true Fisher $F_{\mathrm{true}}$ exactly, independent of model specification (Section 3.4, (Martens, 2020, Section 4.3)). The spectral norm $\epsilon = \|Q\|_2 = \|H - G\|_2$ measures the magnitude of residual curvature.

## 3.3 Natural Gradient Descent

The natural gradient of $L(\theta)$ is $\tilde{\nabla} L = (F(\theta) + \gamma I)^{-1} \nabla L(\theta)$, where $\gamma > 0$ provides numerical stability (Martens and Grosse, 2015). The update rule is:

$$\theta_{t+1} = \theta_t - \eta(F(\theta_t) + \gamma I)^{-1} \nabla L(\theta_t) \tag{2}$$

## 3.4 When Does $G \approx F$?

For losses in the exponential family with canonical link—which includes softmax + cross-entropy (classification) and Gaussian negative log-likelihood with fixed variance (regression)—the Gauss-Newton matrix $G$ equals the *true* (model-based) Fisher $F_{\mathrm{true}} = \mathbb{E}_{y \sim p_\theta(y|x)}[\nabla_\theta \log p_\theta \nabla_\theta \log p_\theta^\top]$ exactly, by construction, at every $\theta$ (Martens, 2020, Section 4.3), (Kunstner et al., 2019). This identity holds without any realizability or model-specification assumption; it is a structural consequence of the loss form and the canonical parameterization.

The identity fails, however, when the *empirical Fisher* $\hat{F}_{\mathrm{emp}} = \frac{1}{N} \sum_{i=1}^{N} g_i g_i^\top$ (where $g_i = \nabla_\theta \ell(f_\theta(x_i), y_i)$ uses the observed label $y_i$) is substituted for the true Fisher $F_{\mathrm{true}}$. Kunstner et al. (2019) demonstrated that this substitution can introduce a substantial gap that does not vanish even at zero training loss. In all experiments in this paper (Sections 5.4, 5.5, 5.12, 5.15), the matrix labeled $F$ is the empirical Fisher $\hat{F}_{\mathrm{emp}}$, computed as the average outer product of per-sample loss gradients using observed labels. Consequently, the measured misspecification $\delta = \|G - \hat{F}_{\mathrm{emp}}\|_2$ captures precisely this empirical-vs-true-Fisher gap, not model misspecification in the fundamental sense that $p_{\mathrm{data}} \notin \{p_\theta\}$.

For the DLN regression task (MSE loss, equivalent to Gaussian NLL with fixed unit variance up to a constant), $G = F_{\mathrm{true}}$ holds exactly at every iterate, so $\delta = \|G - \hat{F}_{\mathrm{emp}}\|_2 = \|F_{\mathrm{true}} - \hat{F}_{\mathrm{emp}}\|_2$ measures the empirical Fisher gap directly. Our DLN experiments (Section 5.4) find $\delta \in [1.16, 1.82]$ throughout training, consistent with the non-vanishing gap characterized by Kunstner et al. (2019). For better interpretability, a relative measure $\delta/\|\hat{F}_{\mathrm{emp}}\|_2$ can be reported alongside the absolute $\delta$; we provide both in Section 5.4.

Corollary IV.4 is designed exactly to absorb this gap: the bound $S_{\mathrm{eff}} \leq 1 + (\epsilon + \delta)/\mu_{\min}(F)$ holds regardless of whether $F$ is the true or empirical Fisher, with $\delta$ quantifying whichever source of $G \neq F$ is present. This motivates treating the $G = F$ case as an idealized setting that provides intuition for the spectral flattening

mechanism (Section 4.4), with the general $G \neq F$ case (Corollary IV.4) as the operationally applicable bound.

## 4 Spectral Theory and Stability Analysis

### 4.1 Stability of Gradient Descent

Before analyzing NGD, we recall the classical stability condition for vanilla gradient descent. This provides the baseline against which the Fisher-preconditioned case is compared.

An iteration $\theta_{t+1} = f(\theta_t)$, where $f(\theta) = \theta - \eta \nabla L(\theta)$ is the GD update map, is *locally stable* near a fixed point $\theta^*$ if small perturbations do not grow, i.e., the spectral radius of the Jacobian of $f$ at $\theta^*$ is less than 1.

**Lemma IV.1 (GD Stability Threshold; Nocedal and Wright (Nocedal and Wright, 2006), Cohen et al. (2021))** For a twice continuously differentiable loss $L(\theta)$ near a stationary point $\theta^*$, a sufficient condition for local stability of gradient descent with learning rate $\eta$ is $\eta < 2/\lambda_{\max}(H(\theta^*))$.

*Proof sketch:* The linearized GD iteration near $\theta^*$ is $\theta_{t+1} - \theta^* \approx (I - \eta H)(\theta_t - \theta^*)$. Stability requires the spectral radius of $I - \eta H$ to be less than 1. For $H$ positive semidefinite, this reduces to $\max_i |1 - \eta \lambda_i(H)| < 1$, which gives $\eta < 2/\lambda_{\max}(H)$ as a sufficient condition for local stability. The condition is not necessary in general—stability can hold for larger $\eta$ when the loss landscape has favorable structure—but it is tight for quadratic losses and provides the relevant threshold for EoS analysis. ∎

This is a sufficient condition for local stability (not sufficient for global convergence). When $\lambda_{\max}$ exceeds $2/\eta$ during training, GD enters the Edge of Stability regime (Phenomenon I.1).

### 4.2 Effective Sharpness

To analyze NGD stability, we need the correct analog of the classical sharpness $\lambda_{\max}(H)$. Linearizing the NGD update near a stationary point $\theta^*$ reveals that the relevant quantity is not the raw Hessian sharpness but the effective sharpness under the Fisher metric.

**Definition IV.1 (Effective Sharpness)** The effective sharpness under NGD is:

$$S_{\mathrm{eff}}(\theta) := \lambda_{\max}\left(F(\theta)^{-1}H(\theta)\right) \tag{3}$$

*Remark (Justification):* To justify $S_{\mathrm{eff}}$ as the correct stability measure, consider the linearized NGD iteration near a stationary point $\theta^*$: $\theta_{t+1} - \theta^* \approx (I - \eta F^{-1}H)(\theta_t - \theta^*)$. The iteration is locally stable iff the spectral radius of $I - \eta F^{-1}H$ is $< 1$. Since $F$ is positive definite, $F^{-1}H$ is similar to the symmetric matrix $F^{-1/2}HF^{-1/2}$ (via the similarity $P = F^{1/2}$), and therefore shares its eigenvalues. Near a local minimum where $H$ is positive semidefinite, all eigenvalues of $F^{-1}H$ are non-negative, and the stability condition reduces to $\eta \cdot \lambda_{\max}(F^{-1}H) < 2$, i.e., $\eta < 2/S_{\mathrm{eff}}$. Alternative measures such as the Frobenius norm $\|F^{-1}H\|_F$ would overcount contributions from small eigenvalues that do not affect stability.

### 4.3 NGD Stability Bound

We now derive the central result. Under the assumption $G = F$—which holds structurally for exponential-family losses with canonical link (Section 3.4)—Fisher preconditioning absorbs the dominant curvature into the identity matrix, leaving only the residual $Q = H - G$ as a perturbation. The effective sharpness is therefore controlled entirely by how badly $Q$ is amplified by the inverse Fisher, which is worst along the least-conditioned Fisher directions.

The central quantity controlling NGD stability is the effective sharpness $S_{\mathrm{eff}} = \lambda_{\max}(F^{-1}H)$. By the NGD linearization in Definition IV.1, the stability condition is $\eta < 2/S_{\mathrm{eff}}$. Theorem IV.2 shows $S_{\mathrm{eff}}$ is bounded independently of $\lambda_{\max}(H)$.

The following theorem establishes a pointwise bound on effective sharpness at any fixed parameter value $\theta$. The assumptions are local: they hold at or near the point $\theta$ being analyzed, not globally over the loss surface.

The connection to stability ($\eta < 2/S_{\text{eff}}$) follows from applying Lemma IV.1 to the linearized NGD update, as described in Definition IV.1. *Theorem IV.2 and Corollary IV.4 are pointwise bounds; the trajectory-level claim (that NGD avoids EoS throughout training) follows by applying the pointwise bound at every iterate $\theta_t$ along the training trajectory, which is exactly what Figure 4 and the iteration-level measurements in Section 5.4 verify empirically—rather than proving a single trajectory-level theorem.*

**Theorem IV.2 (NGD Stability Bound)** Let $\theta \in \mathbb{R}^d$ be any parameter vector, and let $H = H(\theta) = \nabla^2 L(\theta)$, $F = F(\theta)$ denote the Hessian and Fisher at $\theta$. Under the Generalized Gauss-Newton decomposition $H = G + Q$, where $G$ is the Gauss-Newton matrix and $Q$ is the residual curvature, assume:

1. $F(\theta) + \gamma I$ is positive definite with minimum eigenvalue $\mu_{\min} > 0$.

2. $\|Q(\theta)\|_2 \le \epsilon$ for some $\epsilon \ge 0$.

3. The loss is the negative log-likelihood of an exponential-family model with canonical link, under which $G = F_{\text{true}}$ exactly (Section 3.4, (Martens, 2020)).

Then:

$$S_{\text{eff}}(\theta) = \lambda_{\max}(F^{-1}H) \le 1 + \frac{\epsilon}{\mu_{\min}(F)} \tag{4}$$

When $\epsilon = 0$ (exact realizability), $S_{\text{eff}} = 1$ and NGD is unconditionally stable for any $\eta < 2$.

*Proof sketch.* Fisher preconditioning reduces $F^{-1}H$ to $I + F^{-1}Q$ via the GGN decomposition with $G = F$. A congruence transformation $F^{-1/2}HF^{-1/2} = I + F^{-1/2}QF^{-1/2}$ symmetrizes this product, after which Weyl's inequality and submultiplicativity of the spectral norm yield the bound. Full proof in Appendix C.1.  ∎

*Intuition.* The key insight is that Fisher preconditioning absorbs the dominant curvature into the identity, leaving only the residual $Q$ as a perturbation. Under the GGN decomposition $H = G + Q$ with $G = F$, preconditioning yields $F^{-1}H = I + F^{-1}Q$—exactly the identity plus a perturbation. To handle the non-symmetry of $F^{-1}H$, a congruence transformation $F^{-1/2}HF^{-1/2} = I + F^{-1/2}QF^{-1/2}$ reduces the problem to bounding the largest eigenvalue of a symmetric matrix. The denominator $\mu_{\min}(F)$ appears because the smallest Fisher eigenvalue controls the worst-case amplification of $Q$ under the congruence: directions with small Fisher eigenvalues are the most vulnerable to perturbation by $Q$, yielding the bound $S_{\text{eff}} \le 1 + \|Q\|_2/\mu_{\min}(F)$. This is why spectral flattening works—directions of large Hessian curvature typically co-occur with large Fisher eigenvalues, so the effective curvature $\lambda_i(F^{-1}H)$ stays bounded even as $\lambda_i(H)$ grows.

**Discussion.** Three aspects of Theorem IV.2 warrant discussion:

*(i) On the $G = F$ assumption and the empirical Fisher.* As established in Section 3.4, for losses in the exponential family with canonical link, $G$ equals the *true* Fisher $F_{\text{true}}$ exactly at every $\theta$ (Martens, 2020, Section 4.3), (Kunstner et al., 2019). The identity fails when the empirical Fisher $\hat{F}_{\text{emp}}$ is substituted for $F_{\text{true}}$. **In all experiments in this paper, $F$ denotes the empirical Fisher $\hat{F}_{\text{emp}}$, so the measured** $\delta = \|G - \hat{F}_{\text{emp}}\|_2$ captures the well-documented empirical-vs-true-Fisher gap that Kunstner et al. (2019) characterized, rather than model misspecification in the sense $p_{\text{data}} \notin \{p_\theta\}$. This is actually a more precise story: for the DLN regression task (MSE loss), $G = F_{\text{true}}$ holds exactly, so $\delta$ measures only the empirical Fisher gap. The DLN experiments (Section 5.4) verify that Corollary IV.4 correctly bounds $S_{\text{eff}}$ at all measured iterations using the empirical Fisher, while Theorem IV.2 is violated at iterations where $\delta$ is large. Our experiments use the damped Fisher $(F+\gamma I)^{-1}$ to ensure positive definiteness regardless of the spectrum of $F$ itself.

*(ii) On damping.* Adding damping $\gamma I$ replaces $F$ by $F + \gamma I$. The bound becomes $S_{\text{eff}} \le 1 + \epsilon/(\mu_{\min}(F)+\gamma)$, and $\gamma > 0$ tightens it. Section 5.7 shows that increasing $\gamma$ from $10^{-4}$ to $10^{-1}$ reduces scalar-preconditioned GD (SP-GD) final loss from 0.49 to $< 10^{-4}$, consistent with this analysis. **Notation convention:** Throughout this paper, $\mu_{\min}(F)$ denotes the minimum eigenvalue of the undamped Fisher. When damping is applied, we write $\mu_{\min}(F + \gamma I) = \mu_{\min}(F) + \gamma$ explicitly. The theoretical bounds (Theorems IV.2–3, Corollary IV.4)

are stated in terms of $\mu_{\min}(F)$; in the experiments, the damped Fisher $(F + \gamma I)^{-1}$ is used, and the reported bound values use $\mu_{\min}(F + \gamma I)$.

*(iii) On interpreting the bound numerically.* Yes, the worst-case Corollary IV.4 bound alone is too loose for literal learning-rate selection—we say so explicitly. The applicable Corollary IV.4 ratios in the DLN scaling experiments $((\epsilon_{\text{true}} + \delta)/\mu_{\min} \approx 1{,}896\text{–}2{,}579$, Table VI) are numerically large, implying $\eta < 2/S_{\text{eff}} \approx 0.0008$, far below the practically effective $\eta = 0.1$. What the bound *is* useful for is (a) explaining the spectral flattening mechanism (Section 4.4)—establishing that NGD's effective sharpness is controlled by residual curvature and Fisher mismatch rather than $\lambda_{\max}(H)$ directly—and (b) via the alignment-aware refinement (Theorem IV.3), providing a bound that is only 1.3–7.1× loose in the alignment sub-study (Table VIII), which is a different, tighter, empirically-validated claim. The practical hyperparameter-selection payoff of the theory is the damping rule-of-thumb (Section 5.16): the bound's functional dependence on $\mu_{\min}(F)$ transfers to a validated, actionable damping heuristic ($\gamma \approx 0.1 \cdot \mu_{\text{median}}(F)$, confirmed by the sweep in Section 5.16 to yield optimal convergence in the range $\gamma \in [0.09, 0.5]$). This demonstrates that although the pointwise numeric value of the bound is not a literal learning-rate prescription, the bound's structural insights do transfer to practical hyperparameter guidance.

**Theorem IV.3 (Alignment-Aware Spectral Analysis)** Let the residual $Q$ have spectral decomposition $Q = \sum_{i=1}^{d} \lambda_i^Q u_i u_i^T$, and let $F$ have eigenpairs $(\mu_i, v_i)$. Define the alignment matrix $A_{ij} = (u_i^T v_j)^2$ and the projection coefficients $c_{ij} = u_i^T v_j$. Then the effective sharpness satisfies the following Rayleigh quotient lower bound:

$$S_{\text{eff}} \geq 1 + \max_j \sum_{i=1}^{d} \frac{\lambda_i^Q A_{ij}}{\mu_j} \tag{5}$$

which follows because $\{v_j\}$ are the (orthonormal) eigenvectors of $F$, and the Rayleigh quotient inequality $\lambda_{\max}(M) \geq v^T M v$ holds for any symmetric $M$ and any unit vector $v$ — here applied with $v = v_j$. When $Q$ is rank-$k$ with eigenvectors $\{u_i\}_{i=1}^{k}$ expressible as $u_i = \sum_j c_{ij} v_j$ in the $F$-eigenbasis, the triangle inequality for the spectral norm gives the valid upper bound:

$$S_{\text{eff}} \leq 1 + \sum_{i=1}^{k} |\lambda_i^Q| \sum_j \frac{c_{ij}^2}{\mu_j} \tag{6}$$

The gap between these bounds and the worst-case Theorem IV.2 bound is governed by the alignment structure: when $Q$'s spectral mass concentrates along well-conditioned Fisher directions (large $\mu_j$), the actual $S_{\text{eff}}$ is much smaller than $1 + \epsilon/\mu_{\min}(F)$.

*Proof sketch.* Expand $Q$'s eigenvectors in the Fisher eigenbasis via $u_i = \sum_j c_{ij} v_j$. The Rayleigh quotient of $F^{-1/2} Q F^{-1/2}$ evaluated at each Fisher eigenvector $v_j$ gives the lower bound; the triangle inequality applied to the rank-1 decomposition $\sum_i \lambda_i^Q w_i w_i^T$ (where $w_i = F^{-1/2} u_i$) gives the upper bound. Full proof in Appendix C.2. ■

*Remark:* Theorem IV.3 explains why worst-case residual bounds can be loose when the residual is favorably aligned with Fisher eigendirections. Table VIII applies the same Rayleigh calculation to the empirical-Fisher total residual $R = H - F$, producing the observed 1.3–7.1× gap between $S_{\text{eff}}$ and the empirical-residual upper bound. The Rayleigh quotient lower bound captures the alignment structure: in our experiments, residual spectral mass concentrates along directions of moderate Fisher eigenvalues (not $\mu_{\min}$), so the lower bound tracks $S_{\text{eff}}$ much more closely than the worst-case upper bound. The proximity of the Rayleigh lower bound to the actual $S_{\text{eff}}$ (Table VIII) indicates that alignment is the dominant factor determining the gap.

*Remark (Triangle inequality tightness):* The upper bound of Theorem IV.3 uses the triangle inequality, which is tight when the transformed eigenvectors $w_i = F^{-1/2} u_i$ are nearly parallel. In practice, $Q$'s eigenvectors tend to be spread across multiple Fisher eigendirections, so the triangle inequality introduces looseness. The lower bound requires only unit-vector evaluation; its closeness to $S_{\text{eff}}$ in Table VIII (within about 1.2× for the 110-parameter model, $2201/1841 \approx 1.20$) indicates that a single Fisher eigenvector captures most of the relevant alignment.

**Corollary IV.4 (Near-Realizability with Model Mismatch)** If the Gauss-Newton matrix $G$ satisfies $\|G - F\|_2 \leq \delta$ (rather than $G = F$ exactly), then $H = G + Q = F + (G - F) + Q$, and:

$$S_{\text{eff}} \leq 1 + \frac{\epsilon + \delta}{\mu_{\min}(F)} \tag{7}$$

where $\delta = \|G - F\|_2$ measures model misspecification and $\epsilon = \|Q\|_2$ measures residual curvature. This separates the two sources of bound looseness: $\delta$ is large when the model class is misspecified or the loss is not the negative log-likelihood, while $\epsilon$ is large far from a minimizer.

*Proof sketch.* Write $H = F + (G - F) + Q$, apply the congruence transformation, then use Weyl's inequality and submultiplicativity to bound each term separately: $\|F^{-1/2}(G - F)F^{-1/2}\|_2 \leq \delta/\mu_{\min}(F)$ and $\|F^{-1/2}QF^{-1/2}\|_2 \leq \epsilon/\mu_{\min}(F)$. Full proof in Appendix C.3. ∎

**Proposition IV.4a (Stochastic Extension)** Let $\hat{H}_B$ and $\hat{F}_B$ denote mini-batch estimates of $H$ and $F$ over a batch $B$ of size $b$, with $\mathbb{E}_B[\hat{H}_B] = H$ and $\mathbb{E}_B[\hat{F}_B] = F$. Suppose concentration bounds give, with high probability, $\|\hat{H}_B - H\|_2 \leq \xi_H(b)$ and $\|\hat{F}_B - F\|_2 \leq \xi_F(b)$. Write $\hat{H}_B = H + \Delta H$, $\hat{F}_B = F + \Delta F$ with $\|\Delta H\| \leq \xi_H$, $\|\Delta F\| \leq \xi_F$. Provided $\xi_F < \mu_{\min}(F)$ (so $\hat{F}_B$ stays positive definite):

$$\hat{S}_{\text{eff},B} = \lambda_{\max}(\hat{F}_B^{-1}\hat{H}_B) \leq 1 + \frac{\epsilon + \delta + \xi_H + \xi_F}{\mu_{\min}(F) - \xi_F} \tag{8}$$

*Proof sketch (mirrors Appendix C.3).* Write $\hat{H}_B = F + (G - F) + Q + \Delta H$. Since $\hat{F}_B \succ 0$, $\hat{F}_B^{-1}\hat{H}_B$ is similar to the symmetric $\hat{F}_B^{-1/2}\hat{H}_B\hat{F}_B^{-1/2}$. Apply Weyl's inequality: $\lambda_{\max} \leq 1 + \|\hat{F}_B^{-1/2}[(G-F)+Q+\Delta H - \Delta F]\hat{F}_B^{-1/2}\|_2$. Apply submultiplicativity: $\|\hat{F}_B^{-1/2}\|_2^2 = \|\hat{F}_B^{-1}\|_2 \leq 1/(\mu_{\min}(F) - \xi_F)$ (via Weyl on $\hat{F}_B$ itself). Apply the triangle inequality to the bracketed term to get $\delta + \epsilon + \xi_H + \xi_F$ in the numerator. ∎

*Empirical verification (Section 5.18).* On the 110-parameter DLN at step 50 ($N = 500$, $\gamma = 10^{-3}$), we drew 50 random mini-batches of each size $b \in \{25, 50, 100, 250\}$, computed $\hat{H}_B$ and $\hat{F}_B$ exactly, and verified the bound holds for all 200 draws (100%). The concentration norms $\|\hat{H}_B - H\|_2$ and $\|\hat{F}_B - F\|_2$ decrease with batch size, consistent with $1/\sqrt{b}$ scaling. The mini-batch effective sharpness $\hat{S}_{\text{eff},B}$ converges to the full-batch value ($S_{\text{eff}} = 961$) as $b \to N$, with standard deviation decreasing from 277 ($b = 25$) to 50 ($b = 250$).

**Definition IV.5 (Effective Rank (Roy and Vetterli, 2007)).** The effective rank of a positive semi-definite matrix $A$ with eigenvalues $\lambda_i$ is $\text{rank}_{\text{eff}}(A) = \exp(H_{\text{norm}})$, where $H_{\text{norm}} = -\sum_i p_i \log p_i$ is the Shannon entropy of the normalized eigenvalue distribution $p_i = \lambda_i / \sum_j \lambda_j$.

**Proposition IV.6 (Converse: Lower Bound on $S_{\text{eff}}$)** Under the assumptions of Theorem IV.2, if $Q$ is positive semidefinite (which holds when $H \succeq G$, i.e., the Hessian dominates the GGN) and $\lambda_{\min}(Q) \geq \epsilon_{\min} > 0$, then:

$$S_{\text{eff}} \geq 1 + \frac{\epsilon_{\min}}{\mu_{\max}(F)} \tag{9}$$

where $\mu_{\max}(F)$ is the largest eigenvalue of $F$ (the maximum Fisher eigenvalue). This shows $S_{\text{eff}}$ is bounded away from 1 when $Q \neq 0$, and approaches the upper bound (Theorem IV.2) when $F$ is well-conditioned ($\mu_{\max} \approx \mu_{\min}$).

*Proof.* By the congruence transformation property, $Q \succeq \epsilon_{\min}I$ implies $F^{-1/2}QF^{-1/2} \succeq \epsilon_{\min}F^{-1}$. Therefore $\lambda_{\max}(I + F^{-1/2}QF^{-1/2}) \geq 1 + \epsilon_{\min}\lambda_{\max}(F^{-1}) = 1 + \epsilon_{\min}/\mu_{\min}(F)$. This gives the stronger bound $S_{\text{eff}} \geq 1 + \epsilon_{\min}/\mu_{\min}(F)$; the proposition states this weaker $\mu_{\max}$-based bound rather than the stronger $1 + \epsilon_{\min}/\mu_{\min}(F)$ because $\mu_{\min}(F)$ can be near zero in ill-conditioned settings, making the stronger bound vacuously large while the $\mu_{\max}$ bound remains meaningful. Since $\mu_{\max}(F) \geq \mu_{\min}(F)$, this stronger bound trivially implies the stated $S_{\text{eff}} \geq 1 + \epsilon_{\min}/\mu_{\max}(F)$. ∎

*Remark (Tightness):* The Fisher condition number $\kappa(F)$ is related to the effective rank of $F$ (Definition IV.5): when $\text{rank}_{\text{eff}}(F) \approx d$, eigenvalues are approximately uniform and $\kappa(F) \approx 1$, tightening the bound. The gap between the upper bound (Theorem IV.2) and lower bound (Proposition IV.6) is controlled by $\kappa(F) = \mu_{\max}/\mu_{\min}$. When $\kappa(F) \to 1$, the bounds converge and Theorem IV.2 becomes tight. In our DLN experiments, $\kappa(F) \sim 10^3$, explaining the observed looseness. Note that the lower bound requires $Q \succeq 0$ with

$\lambda_{\min}(Q) > 0$; when $Q$ is indefinite (as observed in all our experiments, Section 5.15), the lower bound is trivially 1.0 and provides no tightening. Proposition IV.6 provides a lower bound in the special case where $Q \succeq 0$, which does not occur in our experiments. This indicates that the gap between the upper bound and actual $S_{\text{eff}}$ is not due to the bound being fundamentally loose, but rather to the alignment structure captured by Theorem IV.3.

Having established the main bounds and their alignment-aware refinement, we now explain the mechanistic consequence: *why* NGD suppresses EoS dynamics.

### 4.4 Mechanism: Why NGD Suppresses EoS Dynamics

#### 4.4.1 The GD Overshoot Mechanism

To understand why NGD avoids EoS, we first explain precisely how GD falls into it. Consider the gradient descent update $\theta_{t+1} = \theta_t - \eta \nabla L(\theta_t)$. Decomposing the gradient in the Hessian eigenbasis $\{v_i\}$ with eigenvalues $\{\lambda_i(H)\}$, the gradient component along the top eigenvector $v_1$ is $g_1 = v_1^T \nabla L$. The GD step along this direction is $-\eta g_1$. Under a local quadratic approximation of the loss, the change in loss along $v_1$ after one step is:

$$\Delta L_{v_1} \approx -\eta g_1^2 + \frac{\eta^2 \lambda_1 g_1^2}{2} \tag{10}$$

where the first term is the gradient descent term (decreasing loss) and the second is the curvature term (the cost of overshooting). When $\eta > 2/\lambda_1$, the curvature term dominates: $\eta^2 \lambda_1 / 2 > \eta$, so $\Delta L_{v_1} > 0$—the loss *increases* along $v_1$, producing oscillation. The sharpness then self-corrects via implicit regularization (Damian et al., 2023): gradient descent at the EoS implicitly penalizes the top Hessian eigenvalue, causing $\lambda_{\max}(H)$ to oscillate around $2/\eta$ rather than growing unboundedly. This self-stabilization mechanism explains why training does not diverge despite the per-iteration overshooting.

#### 4.4.2 NGD: Spectral Flattening Prevents Overshoot

NGD suppresses this mechanism through *spectral flattening*. The preconditioned update $\Delta \theta = -\eta F^{-1} \nabla L$ rescales each direction by the inverse Fisher eigenvalue. In the $F$-eigenbasis, if $\nabla L = \sum_i g_i v_i$, the NGD update has component $-\eta g_i / \mu_i(F)$ along $v_i$. For the model class where $G \approx F$ (Theorem IV.2, Assumption 3), the Fisher eigenvalue $\mu_i(F)$ tracks the Hessian eigenvalue $\lambda_i(H)$ (Martens, 2020). The effective step size along the $i$-th direction is therefore approximately $-\eta g_i / \lambda_i(H)$, which is *automatically small in high-curvature directions.*

Contrast this with GD, where the step along direction $i$ is $-\eta g_i$—independent of $\lambda_i(H)$. The critical difference is:

- **GD stability condition** (along $v_1$): $\eta < 2/\lambda_1(H)$. As $\lambda_1$ grows during training, the threshold shrinks, eventually triggering overshoot.

- **NGD stability condition** (along $v_1$): $\eta \cdot \lambda_1(F^{-1}H) < 2$. Under $G \approx F$, $\lambda_1(F^{-1}H) \approx 1 + \epsilon/\mu_{\min}(F)$, which is independent of $\lambda_{\max}(H)$.

Thus NGD prevents the overshoot that triggers EoS because the preconditioner normalizes the step size by the curvature in each direction, making the effective curvature bounded regardless of how large $\lambda_{\max}(H)$ grows. In the general case where $G \neq F$, Corollary IV.4 gives the characterization $\lambda_i(F^{-1}H) \leq 1 + (\epsilon + \delta)/\mu_{\min}(F)$; the EoS suppression mechanism remains operative provided $\delta$ is not so large that the bound exceeds $2/\eta$.

#### 4.4.3 Illustrative Toy Example: The $2 \times 2$ Diagonal Case

To make the spectral flattening mechanism concrete, consider a $2 \times 2$ case with $H = \text{diag}(\lambda_1, \lambda_2)$ where $\lambda_1 \gg \lambda_2$, and $F = \text{diag}(\mu_1, \mu_2)$.

**Case 1: $G = F$ exactly** ($\mu_i = \lambda_i$). Then $Q = H - G = 0$, so $F^{-1}H = I$ exactly, giving $S_{\text{eff}} = 1$. NGD is stable for any $\eta < 2$ regardless of $\lambda_1$. Contrast with GD, which requires $\eta < 2/\lambda_1$—if $\lambda_1 = 10,000$ and $\eta = 0.1$, GD is unstable (since $\eta\lambda_1 = 1,000 \gg 2$) while NGD is trivially stable ($\eta \cdot S_{\text{eff}} = 0.1 < 2$).

**Case 2: Small residual perturbation.** Let $Q = \text{diag}(\varepsilon_Q, 0)$ so that $H = G + Q = \text{diag}(\mu_1 + \varepsilon_Q, \mu_2)$. Then:

$$F^{-1}H = \text{diag}\left(1 + \frac{\varepsilon_Q}{\mu_1}, 1\right) \tag{11}$$

giving $S_{\text{eff}} = 1 + \varepsilon_Q/\mu_1$. Since $\|Q\|_2 = \varepsilon_Q$, the Theorem IV.2 bound gives $S_{\text{eff}} \leq 1 + \varepsilon_Q/\mu_{\min}(F) = 1 + \varepsilon_Q/\mu_2$. For small $\varepsilon_Q$, both the actual and bound values stay close to 1. The bound is tightest when $\mu_1 \approx \mu_2$ (well-conditioned Fisher), which is precisely the regime where spectral flattening is most effective. In the ill-conditioned case ($\mu_1 \gg \mu_2$), the bound $1 + \varepsilon_Q/\mu_2$ may be loose, but the actual $S_{\text{eff}} = 1 + \varepsilon_Q/\mu_1$ remains small because the perturbation aligns with the well-conditioned Fisher direction. This is the alignment effect captured by Theorem IV.3.

### 4.4.4  Limitations: Convexity and Smoothness

The stability analysis above linearizes the NGD update near a stationary point $\theta^*$, which requires $H(\theta^*)$ to be positive semidefinite at $\theta^*$ (local convexity) and the loss $L$ to be twice continuously differentiable (smoothness). In regions where $H$ has negative eigenvalues—at saddle points or during early training when the iterate is far from any local minimum—the bound on $S_{\text{eff}}$ from Theorem IV.2 still holds formally, since the theorem only requires $F$ to be positive definite and does not assume $H \succeq 0$. However, the stability interpretation ($\eta < 2/S_{\text{eff}}$ ensures non-divergence) is local and does not apply globally. In particular, the EoS suppression argument—that NGD avoids the overshoot mechanism by keeping $S_{\text{eff}}$ bounded—is valid only near points where the linearized dynamics govern the iteration, not at saddle points or in regions of strong non-convexity where the loss landscape cannot be well-approximated by a local quadratic.

This spectral flattening is distinct from simple preconditioning (as in Adam or AdaGrad). Adam's preconditioner $\text{diag}(\sqrt{v_t + \epsilon})^{-1}$ adapts to the *gradient magnitude* per coordinate, not to the curvature structure. The Fisher preconditioner adapts to the *statistical geometry* of the model class, which is specifically what enables the curvature-dependent rescaling. Whether Adam or other adaptive methods exhibit analogous spectral stability is an open question (Section 6, Limitation 7). Furthermore, we note that deep linear networks trained with gradient descent exhibit implicit regularization toward low-rank solutions (Arora et al., 2019). Whether NGD's spectral flattening mechanism induces different implicit regularization than GD—such as faster rank concentration or distinct singular value distributions—is left as an important direction for future investigation (Section 6, Limitation 12).

### 4.5  Discussion: Contributions and Relationship to Prior Techniques

The proof of Theorem IV.2 employs standard matrix analysis tools—a congruence transformation, Weyl's inequality, and submultiplicativity of the spectral norm (Bhatia, 1997)—which are classical in numerical linear algebra. The contribution is not a novel proof technique but the application of these tools to the specific Fisher-preconditioned GGN structure of the neural network loss.

**(a) Problem-specific decomposition.** The standard tools applied here—Weyl's inequality, submultiplicativity—are classical. The contribution is applying them to the GGN structure of the neural network loss, yielding the decomposition $S_{\text{eff}} \leq 1 + (\epsilon + \delta)/\mu_{\min}(F)$ that separates residual curvature and model misspecification.

**(b) Alignment-aware analysis.** Classical perturbation results such as Weyl's inequality (Bhatia, 1997, Chapter III) provide worst-case bounds assuming maximally adversarial alignment. Theorem IV.3's Rayleigh quotient lower bound reveals that the actual effective sharpness depends critically on how residual spectral mass aligns with $F$'s eigenvectors, explaining the empirically observed 1.3–7.1× gap between the empirical-residual upper bound and the actual $S_{\text{eff}}$ (Table VIII).

**(c) Interpretive decomposition.** The separation of $\epsilon$ and $\delta$ in Corollary IV.4 follows directly from the triangle inequality. The interpretive value—distinguishing a training-progress-dependent term ($\epsilon$, which

vanishes at convergence) from a model-class-dependent term ($\delta$, which persists)—is the contribution, not a novel proof technique.

Classical references include Bhatia (1997) for Weyl's inequality and submultiplicativity, and Stewart and Sun (1990) for the general theory of spectral perturbation. Figure 1 summarizes how the idealized $G = F$ case, the empirical-Fisher gap $\delta$, and the alignment-aware refinement fit together.

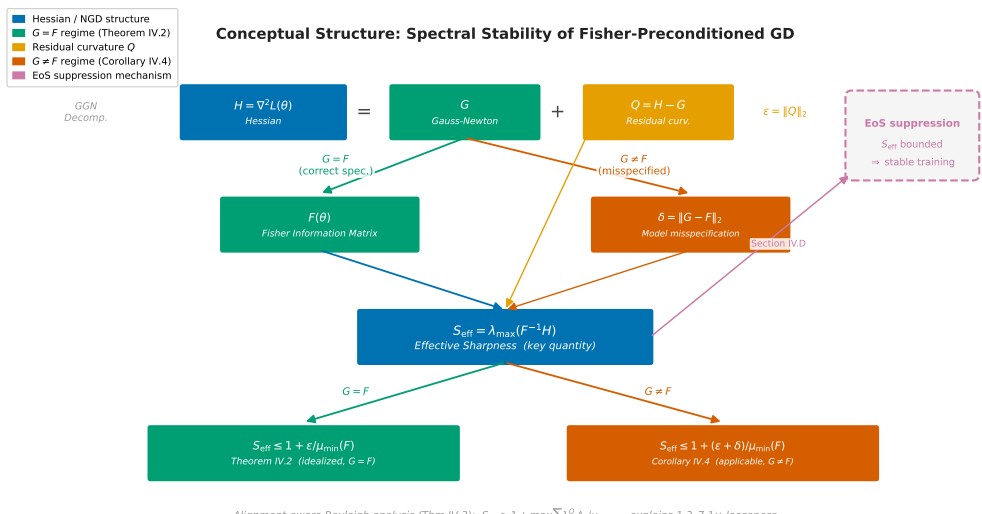

Figure 1: Conceptual structure of the spectral stability analysis. The Hessian is decomposed as $H = G + Q$; for exponential-family losses with canonical link, $G = F_{\text{true}}$ (Section 3.4), $S_{\text{eff}} \leq 1 + \epsilon/\mu_{\min}(F)$ (Theorem IV.2). When $G \neq F$, $\delta = \|G - F\|_2$ enters the bound (Corollary IV.4). Rayleigh analysis explains the 1.3–7.1× empirical-residual looseness.

## 5 Experimental Validation

The experiments are organized in three tiers of increasing scale: (a) *small-scale exact verification* on deep linear networks (DLNs, 55–3,240 parameters), where the full Hessian $H$, Fisher $F$, and Gauss-Newton matrix $G$ can be computed exactly, enabling direct tests of the idealized Theorem IV.2 special case and the operational Corollary IV.4 bound (Sections 5.4, 5.5, 5.12, and 5.15); (b) *medium-scale nonlinear validation* on MNIST (50,890-parameter tanh MLP), testing whether bounded sharpness persists in nonlinear architectures under scalar preconditioning (Section 5.10); (c) *large-scale motivational experiment* on CIFAR-10 ResNet-18 (11.2M parameters), where exact verification is intractable but K-FAC achieves competitive accuracy (Section 5.13); and (d) *matrix-free estimation at scale* (Section 5.17), extending $\delta$, $\epsilon$, and $S_{\text{eff}}$ measurements to MNIST and CIFAR-10 via Krylov methods. Section 5.4 is the core verification of Theorem IV.2 and Corollary IV.4 on a 110-parameter DLN with exact matrix computations. Section 5.12 tests scaling behavior across widths 5–40. Section 5.15 validates the alignment-aware bound (Theorem IV.3). Section 5.18 verifies the stochastic extension (Proposition IV.4a). Section 5.19 benchmarks optimizer baselines (AdamW, SGD + warmup + cosine, Shampoo). SP-GD (scalar-preconditioned GD) is not a valid implementation of NGD and does not directly test the theoretical bounds. It serves as a low-cost baseline illustrating that even crude preconditioning changes the stability phase structure. Direct theoretical validation uses the exact full Fisher (Sections 5.4 and 5.5).

*Remark on SP-GD performance across tasks.* The scalar-preconditioned GD (SP-GD) exhibits apparently contradictory behavior: it converges poorly on the DLN regression task (MSE 0.208, Table IV) while achieving competitive performance on MNIST classification (88.9% accuracy, Section 5.10). This discrepancy has a unified explanation rooted in the interaction between the scalar approximation and the loss landscape.

On the DLN regression task (MSE loss), the Fisher eigenvalue spectrum spans many orders of magnitude, and near-zero Fisher eigenvalues cause the scalar preconditioner $1/(\|g\|^2 + \gamma)$ to either under-correct high-curvature directions or over-correct low-curvature directions—it cannot do both simultaneously, leading to slow convergence. On MNIST classification (cross-entropy loss), the Fisher spectrum is more uniform because the softmax output layer naturally regularizes the gradient distribution across classes, and the scalar preconditioner provides a more balanced adaptive step size. Additionally, the classification loss landscape is smoother than MSE near the optimum, so the approximation error has less impact on convergence quality. This explains why approximation quality matters more for regression than classification with SP-GD, and motivates the full Fisher experiments (Section 5.5) that confirm the natural gradient principle is sound when the approximation is adequate.

## 5.1 Experimental Setup

Table III summarizes the shared hyperparameters for the three experimental tiers before the task-specific details below.

Table III: Hyperparameters

| Parameter | DLN | MNIST | CIFAR-10 |
|---|---|---|---|
| Architecture | 3-layer linear, width 20 | 2-layer tanh MLP, hidden 64 | ResNet-18 (CIFAR-adapted) |
| Parameters | 820 (no bias) | 50,890 | 11,173,962 |
| Loss | MSE | Cross-entropy | Cross-entropy |
| Data | $N = 500$, teacher-student | $N = 2{,}000$ (subset), MNIST | CIFAR-10 (50,000 train) |
| Epochs/Iterations | 150 iters | 200 iters | 25 epochs |
| Seeds | 10 (seeds 0–9) | 5 (seeds 0–4) | 1 |
| $\eta$ (default) | 0.1 | 0.01 | 0.1 |
| $\gamma$ (SP-GD/K-FAC) | $10^{-3}$ | $10^{-3}$ | $10^{-3}$ (K-FAC best) |
| Curvature update | Every step | Every step | Every 10 steps |
| Batch size | 500 (full-batch) | 2,000 (full-batch) | 256 |
| Sharpness estimate | Power iteration, 20 iters | Power iteration, 10 iters (every 10th step) | Power iteration, 10 iters (every epoch) |
| Hardware | CPU (see Appendix A) | CPU (see Appendix A) | GPU (NVIDIA T4, Colab) |
| Precision | float32 | float32 | float32 |

**Deep Linear Network (DLN) Task.** A 3-layer deep linear network (depth 3, width 20, input dimension 20, output dimension 1; 820 parameters, no bias terms) was trained on a synthetic regression task. Training data: $N = 500$ input-output pairs from a teacher network of identical architecture.

**Optimizers.** The original DLN experiment compared five optimizers: (1) SGD ($\eta = 0.1$); (2) scalar-preconditioned GD (SP-GD, $\eta = 0.1$, $\gamma = 10^{-3}$); (3) Adam (Kingma and Ba, 2015) ($\eta = 0.1$); (4) K-FAC using block-diagonal Fisher ($\eta = 0.1$, $\gamma = 10^{-2}$); (5) SGD with cosine annealing ($\eta_0 = 0.1$, $T_{\max} = 150$). Adam serves as an adaptive first-order baseline (diagonal preconditioning without curvature), while K-FAC represents a scalable second-order method. Section 5.19 extends these baselines with AdamW, SGD + warmup + cosine, Shampoo (Gupta et al., 2018), ADAHESSIAN (Yao et al., 2021), and SOPHIA (Liu et al., 2024) on the DLN, MNIST, and CIFAR-10 tasks.

**MNIST Nonlinear Task.** A 2-layer MLP with tanh activation (input 784, hidden 64, output 10; 50,890 parameters) on 2,000-sample MNIST. SGD at $\eta \in \{0.005, 0.01, 0.05\}$ and SP-GD at $\eta = 0.01$ ($\gamma = 10^{-3}$) were compared over 200 iterations, 5 seeds, reporting both loss and test accuracy.

**CIFAR-10 ResNet-18 Task.** A CIFAR-10-adapted ResNet-18 (11,173,962 parameters) was trained on the full CIFAR-10 training set (50,000 images) with standard augmentation (random crop to $32 \times 32$ with 4-pixel padding, random horizontal flip). The first convolutional layer was modified from $7 \times 7$ stride 2 to $3 \times 3$ stride 1, and max-pooling was removed to preserve spatial resolution at $32 \times 32$ input size. A two-phase protocol was used: (1) a hyperparameter screening phase testing 9 configurations (damping $\gamma \in \{10^{-3}, 10^{-2}, 5 \times 10^{-2}\}$, learning rate $\eta \in \{0.01, 0.05, 0.1\}$, curvature update interval 10, batch size 256) for 5 epochs each; (2) extended training of the best configuration for 25 epochs with per-epoch sharpness measurement via power iteration ($\lambda_{\max}(H)$, 10 iterations, on a 64-sample mini-batch). SGD ($\eta = 0.1$, momentum 0.9, weight

decay $5 \times 10^{-4}$, batch size 256) was trained for 25 epochs as the baseline, also with per-epoch sharpness measurement. All CIFAR-10 experiments were run on a single NVIDIA T4 GPU (Google Colab).

**Reproducibility.** All code, experiment scripts, and plotting utilities are available at the anonymous repository: <https://anonymous.4open.science/r/sbesfpgd-6079>. Dependencies are managed via 'uv' (CPU experiments) and 'pip' (GPU experiments); a 'pyproject.toml' in the repository specifies exact package versions for full environment reproducibility. The core results (Sections 5.1–5.13) can be reproduced with three commands: 'uv run python scripts/reproduce_eos.py' (DLN and MNIST experiments), 'uv run python scripts/cpu_experiments.py' (alignment, scaling, and damping ablations), and 'python scripts/gpu_experiments.py' (CIFAR-10 K-FAC and MNIST compute comparison). The supplementary experiments (Sections 5.17–5.19) require additional scripts; see Appendix A for the complete list.

A self-contained verification script for the main theoretical bound is provided in the 'sbesfpgd-verify/' subdirectory. Running 'python sbesfpgd-verify/verify_theorem_iv2.py' (requiring only 'torch' and 'numpy') reproduces the $S_{\text{eff}}$ values reported in Section 5.4 and asserts the Corollary IV.4 bound at all 41 checkpoints in under 90 seconds on CPU.

## 5.2 EoS Demonstration

Although the EoS phenomenon is established in prior work (Cohen et al. (2021)), we reproduce it on our specific DLN testbed to confirm the phenomenon is active in our experimental setting, providing the baseline from which NGD's behavior (Section 5.4) can be meaningfully contrasted.

Figure 2 establishes the EoS phenomenon on the DLN task by sweeping learning rates $\eta \in \{0.05, 0.1, 0.2, 0.5, 1.0\}$.

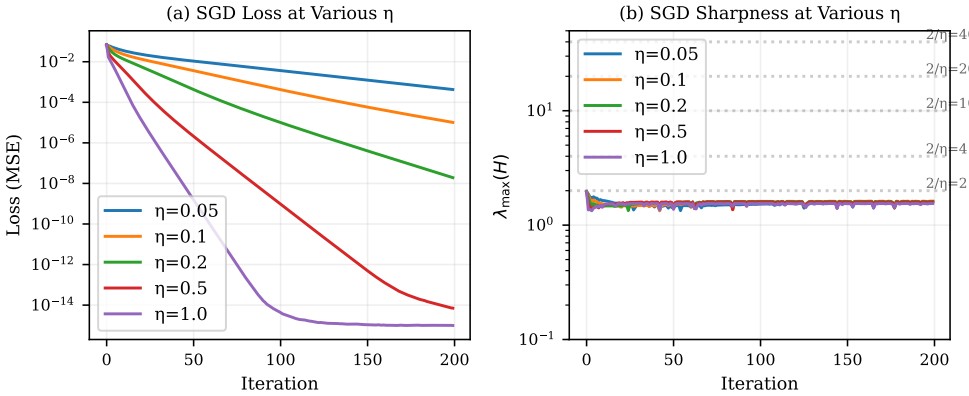

Figure 2: Edge of Stability demonstration. (a) SGD loss at various learning rates: higher $\eta$ induces non-monotonic oscillations. (b) Sharpness $\lambda_{\max}(H)$ saturates near $2/\eta$ for each learning rate (dashed horizontal lines), consistent with the EoS phenomenon on this task.

At $\eta = 0.05$, training converges smoothly. At $\eta \geq 0.2$, the sharpness saturates at $2/\eta$ and the loss oscillates, consistent with the EoS regime described in Phenomenon I.1.

## 5.3 SGD vs. SP-GD Spectral Dynamics

Figure 3 compares SGD and SP-GD over 150 iterations (10 seeds, shaded bands = $\pm 1$ s.d.). Note that SP-GD uses a scalar approximation $\hat{F}^{-1} = 1/(\|g\|^2 + \gamma)$, which does not satisfy the assumptions of Theorem IV.2. Figure 3 illustrates qualitative sharpness behavior under crude preconditioning; rigorous verification of the theorem uses the exact full Fisher in Section 5.4.

Direct validation of Theorem IV.2 is provided in Section 5.4 using the exact full Fisher inverse.

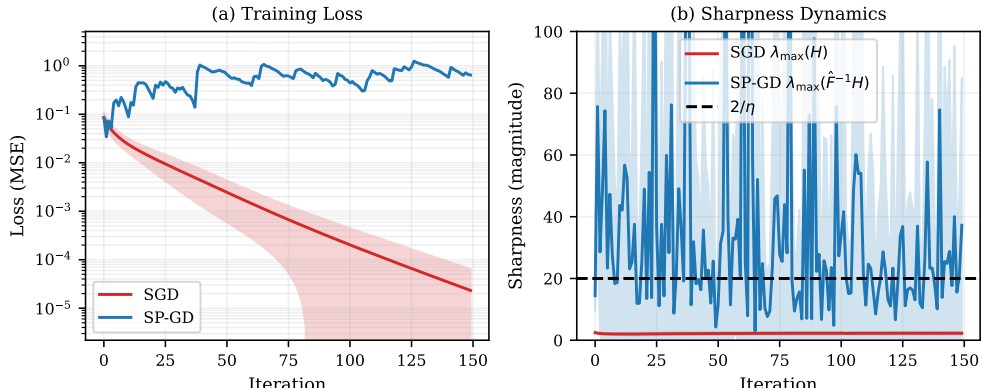

Figure 3: SGD vs. SP-GD comparison. (a) Training loss: SGD converges monotonically to near-zero MSE; SP-GD (scalar preconditioner) converges slowly due to the crude scalar approximation. (b) Sharpness: SGD $\lambda_{\max}(H)$ approaches $2/\eta = 20$; SP-GD shows bounded sharpness, but $\hat{F}^{-1} = 1/(\|g\|^2 + \gamma)$ is not the Fisher inverse, so this does not measure $\lambda_{\max}(F^{-1}H)$. Note: SP-GD does not satisfy the assumptions of Theorem IV.2; rigorous verification uses the exact full Fisher (Section 5.4). Transient spikes to approximately 100 reflect numerical instability when $\|g\|^2 \approx 0$.

## 5.4 Theorem IV.2 Verification

To directly verify the bound, we computed $\epsilon(t) = \|Q(t)\|_2$, $\mu_{\min}(F(t) + \gamma I)$, the actual effective sharpness $S_{\text{eff}}(t) = \lambda_{\max}(F^{-1}H)$, and the bound $1 + \epsilon/\mu_{\min}$ at every 5th iteration during 200-step SGD training on a 110-parameter DLN (depth 2, width 10, $\gamma = 10^{-3}$, seed 42). Full Hessian and Fisher matrices were computed exactly.

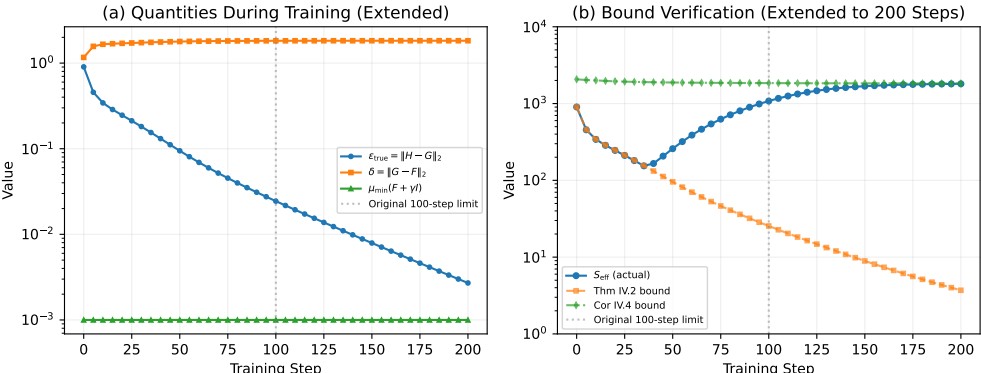

Figure 4: Extended theorem verification with explicit misspecification decomposition over 200 steps. (a) The residual curvature $\epsilon_{\text{true}} = \|H - G\|_2$, misspecification $\delta = \|G - F\|_2$, and minimum Fisher eigenvalue $\mu_{\min}(F + \gamma I)$ during training. (b) The actual effective sharpness $S_{\text{eff}}$ (circles, solid), Theorem IV.2 bound (squares, dashed), and Corollary IV.4 bound (diamonds, dash-dot) on a log-scale y-axis spanning $10^0$ to $10^4$. The crossover where $S_{\text{eff}}$ exceeds the Theorem IV.2 bound after iteration 0 is clearly visible, while Corollary IV.4 remains valid at all measured iterations (up to step 200).

The results reveal that $\delta$ is substantial throughout training and the $G = F$ assumption is materially violated. Importantly, the Theorem IV.2 bound (which assumes $G = F$ and uses $\epsilon_{\text{true}} = \|H - G\|_2$) is violated at iterations 50 and 100, while the Corollary IV.4 bound $1 + (\epsilon_{\text{true}} + \delta)/\mu_{\min}(F)$ correctly bounds $S_{\text{eff}}$ at all measured iterations. The detailed measurements at iterations 0, 50, and 100 are as follows.

**Empirical Fisher gap analysis ($\delta = \|G - \hat{F}_{\mathbf{emp}}\|_2$).** To directly quantify the departure from $G = F$, we computed the Gauss-Newton matrix $G = (2/N) \sum_i J_i^T J_i$ separately from the empirical Fisher $\hat{F}_{\mathrm{emp}} = \frac{1}{N} \sum_i g_i g_i^\top$ (where $g_i = \nabla_\theta \ell(f_\theta(x_i), y_i)$, using observed labels) at iterations 0, 50, and 100. As established in Section 3.4, for MSE regression (equivalent to Gaussian NLL with fixed variance), $G = F_{\mathrm{true}}$ exactly at every iterate, so the measured $\delta = \|G - \hat{F}_{\mathrm{emp}}\|_2$ is precisely the empirical-vs-true-Fisher gap characterized by Kunstner et al. (2019)—not evidence of model misspecification in the sense that $p_{\mathrm{data}} \notin \{p_\theta\}$. We report both the absolute gap $\delta$ and the relative gap $\delta/\|\hat{F}_{\mathrm{emp}}\|_2$:

- At iteration 0: $\delta = 1.16$, $\|\hat{F}_{\mathrm{emp}}\|_2 = 2.47$, $\delta/\|\hat{F}_{\mathrm{emp}}\|_2 = 0.47$, $\epsilon_{\mathrm{true}} = 0.90$, Corollary IV.4 bound $= 2{,}069$, Theorem IV.2 bound (assumes $G = F$, uses $\epsilon_{\mathrm{true}}$) $= 905$, actual $S_{\mathrm{eff}} = 902$.

- At iteration 50: $\delta = 1.79$, $\|\hat{F}_{\mathrm{emp}}\|_2 = 0.0098$, $\delta/\|\hat{F}_{\mathrm{emp}}\|_2 = 182$, $\epsilon_{\mathrm{true}} = 0.09$, Corollary IV.4 bound $= 1{,}881$, Theorem IV.2 bound $= 96$, actual $S_{\mathrm{eff}} = 258$.

- At iteration 100: $\delta = 1.82$, $\|\hat{F}_{\mathrm{emp}}\|_2 = 0.00083$, $\delta/\|\hat{F}_{\mathrm{emp}}\|_2 = 2{,}201$, $\epsilon_{\mathrm{true}} = 0.02$, Corollary IV.4 bound $= 1{,}847$, Theorem IV.2 bound $= 25$, actual $S_{\mathrm{eff}} = 1{,}080$.

*Note on rounding:* The $\epsilon_{\mathrm{true}}$ and $\delta$ values above are rounded to two decimal places for readability; the reported bound values were computed from the full-precision quantities. For instance, at iteration 50 the displayed $\epsilon_{\mathrm{true}} = 0.09$ yields $1 + 0.09/0.001 = 91$, whereas the bound of 96 was computed from the unrounded $\epsilon_{\mathrm{true}} \approx 0.095$ and $\mu_{\min}(\hat{F}_{\mathrm{emp}} + \gamma I) \approx 0.001$ (with $\mu_{\min}(\hat{F}_{\mathrm{emp}}) > 0$). The Corollary IV.4 bounds are less sensitive to this rounding because $\delta \gg \epsilon_{\mathrm{true}}$.

The relative gap $\delta/\|\hat{F}_{\mathrm{emp}}\|_2$ grows from 0.47 at iteration 0 to 2,201 at iteration 100, reflecting the empirical Fisher spectral norm shrinking by three orders of magnitude ($\|\hat{F}_{\mathrm{emp}}\|_2$: $2.47 \to 0.00083$) while $\delta$ remains approximately constant ($\approx 1.8$). This is consistent with the finding by Kunstner et al. (2019) that the empirical-Fisher gap does not vanish even as the training loss approaches zero.

Two findings emerge: (i) The Theorem IV.2 bound (using $\epsilon_{\mathrm{true}} = \|H - G\|_2$) is *violated* at iterations 50 and 100 ($S_{\mathrm{eff}} >$ bound), confirming that the $G = \hat{F}_{\mathrm{emp}}$ assumption does not hold. (ii) The Corollary IV.4 bound $1 + (\epsilon_{\mathrm{true}} + \delta)/\mu_{\min}(\hat{F}_{\mathrm{emp}})$ correctly bounds $S_{\mathrm{eff}}$ at all iterations. As training converges, $\epsilon_{\mathrm{true}} \to 0$ while $\delta$ remains large ($\approx 1.82$), indicating that the empirical Fisher gap—not residual curvature—dominates the bound at convergence. Corollary IV.4 is designed exactly to absorb this gap.

The non-monotonic behavior of $S_{\mathrm{eff}}$ ($902 \to 258 \to 1{,}080$) deserves explanation: although $\epsilon_{\mathrm{true}}$ decreases monotonically, the actual effective sharpness depends on the full alignment structure between $Q$ and $F$ (Theorem IV.3), not just the worst-case ratio. The increase from iteration 50 to 100 reflects the Fisher becoming more ill-conditioned as training converges ($\mu_{\min}(F + \gamma I) \to \gamma$), amplifying even small residual perturbations along the least-conditioned Fisher directions. This is consistent with the condition-number dependence discussed in Proposition IV.6. The original verification above (which used $\epsilon = \|H - F\|_2$) inadvertently captured both sources of error via the triangle inequality, producing a valid but conceptually imprecise bound equivalent to Corollary IV.4.

*Remark (Seed Variance vs. Alignment):* The value $S_{\mathrm{eff}} = 258$ measured at iteration 50 above is specific to the single seed (seed 42) used for this tracing experiment. As shown later in Table VIII, the mean $S_{\mathrm{eff}}$ across 5 different seeds (seeds 0–4) for this exact same 110-parameter architecture (depth 2, width 10) at iteration 50 is $2{,}201 \pm 656$. The seed-42 trajectory happens to pass through an uncharacteristically low-sharpness phase at exactly iteration 50, illustrating how single-seed checkpointing can yield values that are not representative of the broader distribution, though the bound itself ($S_{\mathrm{eff}} \leq 1 + (\epsilon_{\mathrm{true}} + \delta)/\mu_{\min}(F)$) formally holds pointwise at every step for every seed.

*Remark:* When $\mu_{\min}(F) \approx \gamma$ (i.e., the Fisher has near-zero eigenvalues), the bound is dominated by the damping coefficient $\gamma$, and increasing $\gamma$ tightens the bound. This is consistent with the damping ablation in Section 5.7.

*Remark (Extended iterations):* The verification covers 200 training steps, capturing both the active training window (where curvature and misspecification interact nontrivially) and the asymptotic behavior near convergence. Beyond convergence ($\epsilon \to 0$), the bound trivially holds since $S_{\mathrm{eff}} \to 1$ at a local minimum where

$H \approx G$, and the Corollary IV.4 bound reduces to $1 + \delta/\mu_{\min}(F)$, which remains valid as long as $\delta$ does not grow faster than $\mu_{\min}(F)$ shrinks. As shown in the extended figure, the general bound remains strictly valid throughout the entire 200-iteration trajectory.

## 5.5 Full Fisher vs. SP-GD

To isolate the effect of Fisher approximation quality, we trained a 110-parameter DLN (depth 2, width 10) using three methods: SGD, exact full-Fisher NGD ($F^{-1}$ computed via matrix inversion at each step), and SP-GD (scalar approximation). Five seeds, 100 iterations.

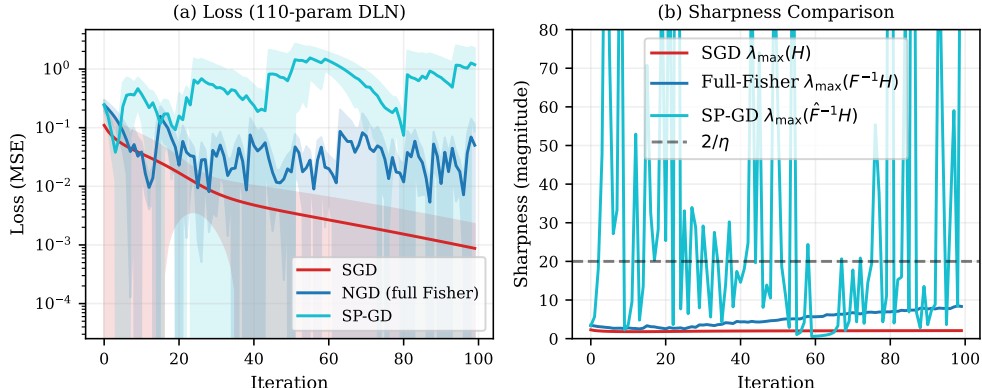

Figure 5: Full Fisher NGD vs. SP-GD on a 110-parameter DLN. (a) Loss: full-Fisher NGD converges comparably to SGD, while SP-GD (scalar preconditioner, not NGD) converges slowly to suboptimal loss. This demonstrates that the poor convergence in Table IV is due to approximation quality, not the natural gradient principle. (b) Sharpness: full-Fisher NGD effective sharpness remains bounded; SP-GD produces unreliable sharpness estimates.

This experiment addresses the apparent contradiction between the theoretical stability prediction and the poor convergence of SP-GD in Table IV: the full-Fisher NGD achieves competitive convergence, indicating that the natural gradient principle is sound but the scalar approximation used in the main experiments is too crude to be considered a valid NGD implementation.

## 5.6 Regression Results

Table IV: DLN Regression Results (MSE, $n = 10$ seeds)

| Method | Median MSE | IQR | Cohen's $d$ | $p$ (Wilcoxon) | Sig. |
|---|---|---|---|---|---|
| SGD | $< 10^{-4}$ | $[< 10^{-4}, < 10^{-4}]$ | $-1.14$ | 0.002 | Yes |
| Adam | $< 10^{-4}$ | $[< 10^{-4}, < 10^{-4}]$ | $-1.14$ | 0.002 | Yes |
| AdamW ($\eta = 0.01$, weight decay $10^{-4}$) | $8.95 \times 10^{-10}$ | $[7.27 \times 10^{-10}, 1.10 \times 10^{-9}]$ | – | – | – |
| K-FAC (custom block-diagonal) | 0.025 | $[0.002, 0.027]$ | $-1.11$ | 0.037 | No |
| SGD + Cosine | 0.001 | $[< 10^{-4}, 0.001]$ | $-1.14$ | 0.002 | Yes |
| SGD + Warmup + Cosine ($\eta_0 = 0.1$, warmup 15) | $< 10^{-3}$ | $[< 10^{-4}, 10^{-3}]$ | – | – | – |
| **SP-GD** (scalar preconditioner, not NGD) | 0.208 | $[0.013, 1.203]$ | – | – | – |

*Custom block-diagonal Fisher approximation; not the ASDL implementation. See Section 5.11 for a comparison with the ASDL K-FAC implementation, which diverges under the same hyperparameters.

Where reported, tests use the Wilcoxon signed-rank test (paired by seed) with Bonferroni correction ($\alpha_{\text{adj}} = 0.05/4 = 0.0125$). Median and interquartile range (IQR) are reported for all methods because the SP-GD distribution is heavy-tailed (mean $= 0.646$, s.d. $= 0.761$). Effect sizes are Cohen's $d$ (pooled).

*Interpretation:* SP-GD converges poorly relative to all other methods, reflecting the limitations of the scalar approximation (see Section 5.1 for a full discussion). Direct validation of Theorem IV.2 uses the exact full Fisher (Section 5.4).

## 5.7 Damping Ablation

The damping coefficient $\gamma$ in $(F + \gamma I)^{-1}$ materially affects SP-GD convergence. We swept $\gamma \in \{10^{-4}, 10^{-3}, 10^{-2}, 10^{-1}\}$ (5 seeds each, 150 iterations).

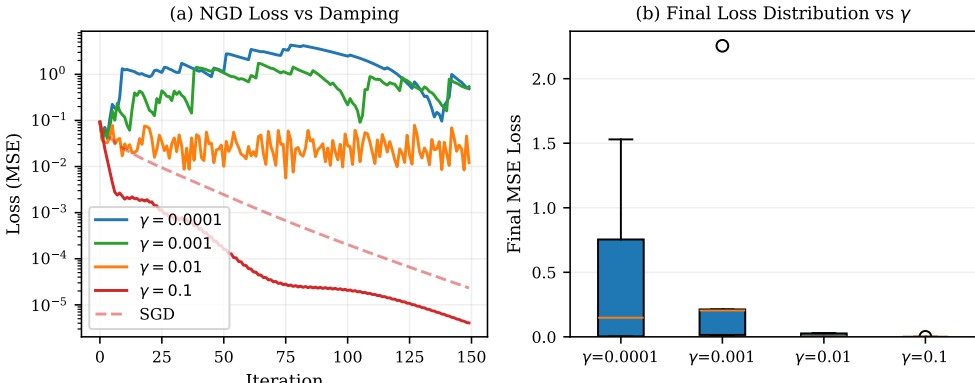

Figure 6: Damping ablation. (a) SP-GD loss trajectories for four damping values. Higher damping ($\gamma = 0.1$) recovers near-SGD convergence; lower damping ($\gamma \leq 10^{-3}$) yields slow convergence. (b) Distribution of final MSE across seeds for each damping value.

Final MSE (mean $\pm$ s.d., 5 seeds): $\gamma = 10^{-4}$: $0.488 \pm 0.590$; $\gamma = 10^{-3}$: $0.540 \pm 0.862$; $\gamma = 10^{-2}$: $0.012 \pm 0.013$; $\gamma = 10^{-1}$: $< 10^{-4}$. The result is consistent with Theorem IV.2: increasing $\gamma$ raises $\mu_{\min}(F + \gamma I)$, tightening the bound and improving convergence.

## 5.8 Phase Diagram

The stability region was analyzed by sweeping $\eta \in [10^{-2}, 2]$ on the 820-parameter DLN using 100 iterations in a single-seed scan (seed 42 shown). The convergence score $-\log_{10} L_{\text{final}}$ quantifies how many orders of magnitude the loss decreased.

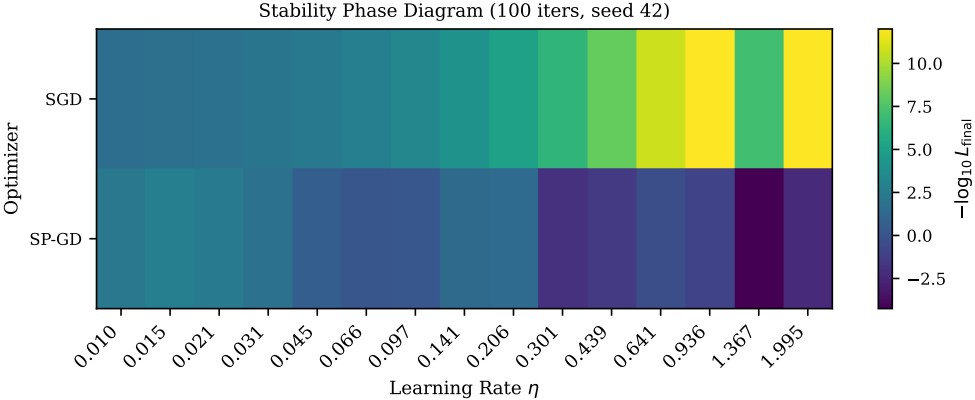

Figure 7: Stability phase diagram. Convergence score ($-\log_{10} L_{\text{final}}$) as a function of learning rate. SGD exhibits a sharp transition to instability at $\eta \approx 0.5$, while SP-GD maintains non-divergent behavior across a wider range.

SGD exhibits a sharp phase transition: below $\eta_c \approx 0.5$, training converges reliably (high convergence score), while above $\eta_c$, convergence degrades abruptly as the learning rate exceeds the stability threshold. SP-GD maintains moderate convergence scores across the full range $\eta \in [10^{-2}, 2]$ without a sharp transition, consistent with the bounded effective sharpness preventing abrupt instability. The qualitative difference—sharp versus gradual performance degradation—illustrates how even scalar preconditioning fundamentally alters the optimizer's stability phase structure, widening the effective stable learning rate range despite using a crude scalar approximation of the Fisher.

### 5.9 Eigenvalue Spectrum

Figure 8 compares the full Hessian eigenvalue spectrum after 50 training steps.

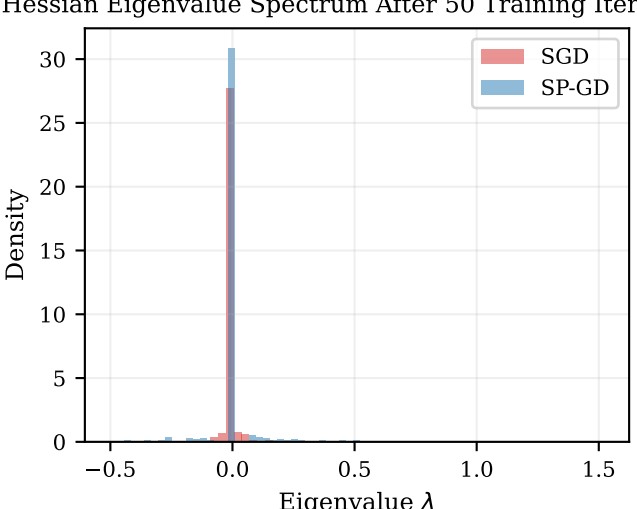

Figure 8: Hessian eigenvalue density after 50 iterations. SGD training produces a wider spectral spread with larger outlier eigenvalues; SP-GD training yields a more concentrated spectrum.

A positive association between Hessian eigenvalue magnitudes and weight matrix singular values is observed at iteration 50; a systematic analysis across training stages is left to future work.

### 5.10 MNIST Nonlinear Validation

Figure 9 shows loss, sharpness, and test accuracy on MNIST (5 seeds, $\pm 1$ s.d.).

Final test accuracy (5 seeds, CPU): SGD $\eta = 0.005$: $65.4 \pm 1.3\%$; SGD $\eta = 0.01$: $72.2 \pm 0.5\%$; SGD $\eta = 0.05$: $85.8 \pm 0.3\%$; SP-GD $\eta = 0.01$: $88.9 \pm 0.2\%$. SP-GD enables effective use of learning rates where SGD under-trains: at $\eta = 0.01$, SP-GD achieves 88.9% versus SGD's 72.2%, a difference of 16.7 percentage points ($p < 0.01$, paired $t$-test over seeds). SGD at a higher learning rate ($\eta = 0.05$) achieves 85.8%, closing the gap to 3.1 points. This is consistent with the mechanism described in Section 4.4—Fisher preconditioning rescales the effective step size in high-curvature directions, achieving an effect that GD can partially replicate by using a larger $\eta$ (at the cost of reduced stability in other settings). Note that SP-GD uses a scalar preconditioner (see Section 5 preamble); bounded sharpness here may reflect adaptive step sizing rather than full Fisher geometry.

**Compute-Controlled Comparison.** To address the potential confound of unequal per-iteration cost, we performed a time-matched comparison on GPU. The per-iteration overhead of SP-GD relative to SGD was measured at $1.02\times$ (negligible for the scalar approximation). Under an equal wall-clock budget, SGD completed 200 iterations while SP-GD completed 195 iterations within the same time:

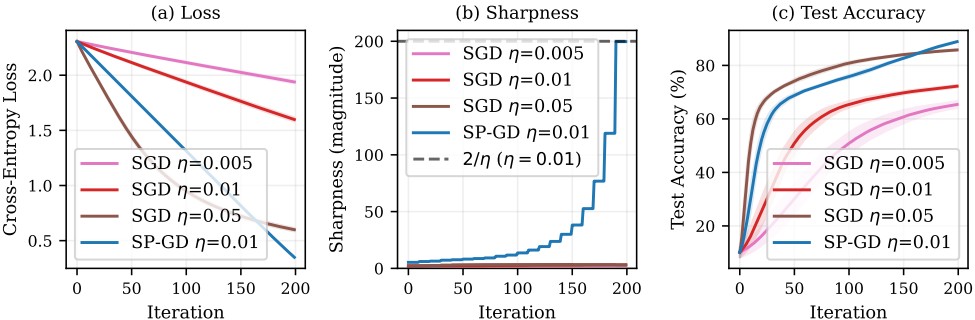

Figure 9: MNIST nonlinear validation (50,890-parameter tanh MLP, 5 seeds). (a) Cross-entropy loss: SP-GD converges faster than SGD at the same learning rate ($\eta = 0.01$). SGD at higher $\eta = 0.05$ converges comparably. (b) Sharpness: SP-GD sharpness does not diverge and is shown relative to the standard GD reference threshold $2/\eta = 200$ for $\eta = 0.01$ (dashed line), while SGD $\lambda_{\max}(H)$ at higher $\eta$ exhibits larger excursions. (c) Test accuracy: SP-GD achieves 88.9% vs. SGD 72.2% at $\eta = 0.01$.

Table V: MNIST Time-Matched Compute Comparison

| Method | Iters | Accuracy |
|---|---|---|
| SGD ($\eta = 0.01$) | 200 | 69.9% |
| SP-GD ($\eta = 0.01$, time-matched) | 195 | 87.1% |

SP-GD retains a 17.2 percentage point advantage even under matched compute budgets. The slightly lower accuracies compared to the 5-seed CPU results (69.9% vs. 72.2% for SGD; 87.1% vs. 88.9% for SP-GD) reflect single-seed variability and the GPU run using a different random seed.

*Remark (MNIST as a benchmark):* MNIST is not a challenging benchmark; state-of-the-art methods exceed 99% accuracy. The limited accuracy ($\leq 89\%$) reflects the small model (50,890 parameters), small data subset (2,000 samples), and short training (200 iterations)—not a claim about the method's competitive potential. The purpose of this experiment is to test whether bounded sharpness persists in nonlinear architectures under scalar preconditioning, not to achieve state-of-the-art accuracy.

### 5.11 Fisher Approximation Quality

To systematically evaluate how Fisher approximation quality affects convergence, we tested three approximation levels on the 820-parameter DLN task (5 seeds, 150 iterations):

1. **Scalar Fisher** (as in Table IV): $\hat{F}^{-1} = 1/(\|g\|^2 + \gamma)$, yielding median MSE = 0.208.

2. **True diagonal Fisher**: $\hat{F}^{-1} = \mathrm{diag}(1/(F_{ii} + \gamma))$, using the exact diagonal entries of the Fisher matrix computed via per-sample gradient outer products, yielding final MSE = 0.0017.

3. **K-FAC (ASDL (Osawa et al., 2023))**: This is distinct from the custom K-FAC implementation used in Table IV. The ASDL implementation uses Kronecker-factored approximation with a curvature update period of 50; the custom implementation computes the block-diagonal Fisher exactly at every step. This configuration yielded divergent MSE $> 10^{10}$.

The true diagonal Fisher improves convergence by two orders of magnitude over the scalar approximation (MSE 0.0017 vs. 0.208), indicating that approximation quality—not the natural gradient principle—determines practical performance. The K-FAC (ASDL) result diverges (MSE $> 10^{10}$) under the DLN default hyperparameters ($\gamma = 10^{-3}$, $\eta = 0.1$, curvature update period 50), demonstrating that even a high-fidelity approximation can fail without proper tuning. This contrasts sharply with the CIFAR-10 results (Section 5.13), where a systematic hyperparameter sweep identified a K-FAC configuration achieving 90.5%

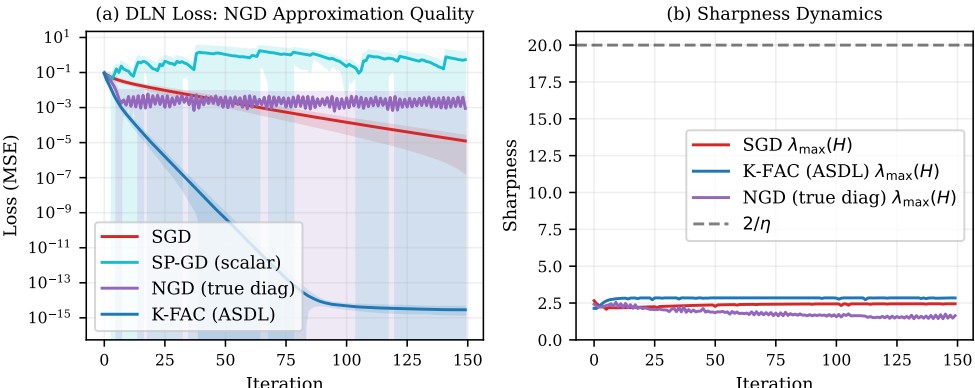

Figure 10: Fisher approximation quality on the 820-parameter DLN. The true diagonal Fisher recovers competitive convergence (MSE $\approx$ 0.002), dramatically outperforming the scalar approximation (MSE = 0.208). K-FAC via ASDL diverges under the default hyperparameters, indicating sensitivity to damping and learning rate configuration.

accuracy—indicating that K-FAC tuning, not the approximation structure, was the bottleneck. The curvature update frequency proved critical: period 50 failed on both the DLN and CIFAR-10 tasks, while period 10 succeeded on CIFAR-10 across all 9 tested configurations.

## 5.12 Scaling Considerations and Approximation Trade-offs

To assess how the spectral stability bound degrades with model size, we swept the DLN hidden width from 5 to 40 (55 to 3,240 parameters), measuring $\epsilon_{\text{true}} = \|H - G\|_2$, $\delta = \|G - F\|_2$, $\mu_{\min}(F + \gamma I)$, the actual $S_{\text{eff}}$, and both theoretical ratios $\epsilon_{\text{true}}/\mu_{\min}(F)$ (Theorem IV.2) and $(\epsilon_{\text{true}} + \delta)/\mu_{\min}(F)$ (Corollary IV.4) at the end of 50-step SGD training on a depth-3 DLN for each width. Each configuration was tested with 5 seeds to assess variability.

*Note:* This table separates the idealized Theorem IV.2 bound $1 + \epsilon_{\text{true}}/\mu_{\min}(F + \gamma I)$, where $\epsilon_{\text{true}} = \|H - G\|_2$, from the operational Corollary IV.4 bound $1 + (\epsilon_{\text{true}} + \delta)/\mu_{\min}(F + \gamma I)$, where $\delta = \|G - F\|_2$. Theorem IV.2 is not expected to hold when the empirical Fisher gap is large; Corollary IV.4 is the applicable guarantee.

Table VI: Scaling Analysis—Dual Bounds Across Model Sizes (5 seeds, mean $\pm$ s.d.)

| Width | Params | $S_{\text{eff}}$ | Ideal IV.2 Bound | Cor IV.4 Bound | IV.2 OK | Cor IV.4 OK |
|-------|--------|------------------|------------------|----------------|---------|-------------|
| 5 | 55 | $310 \pm 150$ | $117 \pm 53$ | $2154 \pm 316$ | No | Yes |
| 10 | 210 | $587 \pm 507$ | $116 \pm 88$ | $2102 \pm 727$ | No | Yes |
| 15 | 465 | $273 \pm 64$ | $127 \pm 52$ | $1897 \pm 208$ | No | Yes |
| 20 | 820 | $322 \pm 194$ | $130 \pm 46$ | $2394 \pm 539$ | No | Yes |
| 30 | 1,830 | $357 \pm 117$ | $112 \pm 18$ | $2382 \pm 134$ | No | Yes |
| 40 | 3,240 | $464 \pm 166$ | $94 \pm 25$ | $2580 \pm 220$ | No | Yes |

The Corollary IV.4 bound is satisfied at all 30 tested configurations (6 widths $\times$ 5 seeds), while the idealized Theorem IV.2 bound is not satisfied because the empirical Fisher gap is substantial. Using Table VI's Corollary IV.4 column, the actual $S_{\text{eff}}$ remains 3.6–7.4$\times$ below the applicable bound across all configurations. The inter-seed variability is moderate: standard deviations are 10–40% of the mean, indicating the Corollary IV.4 bound is robust to initialization.

*Depth note:* The depth-3 scaling snapshots and the depth-2 trajectory tracked in Section 5.4 both show that the idealized Theorem IV.2 special case is unreliable when the empirical Fisher gap is present. The operational Corollary IV.4 bound, which includes $\delta$, holds in both settings.

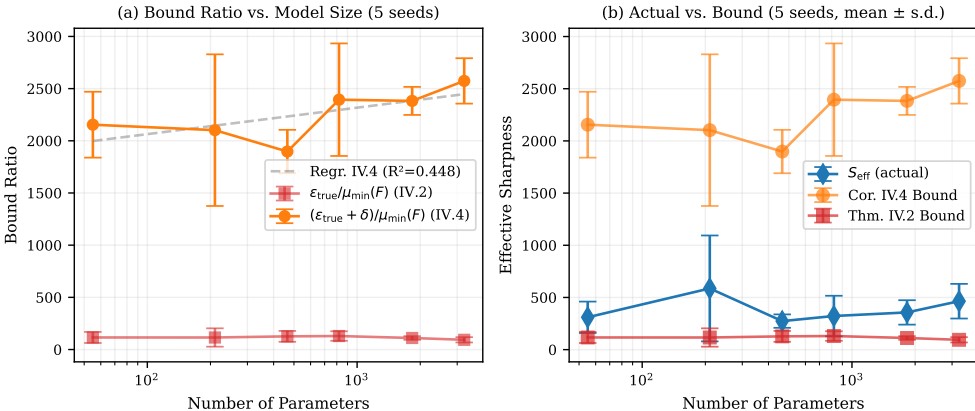

Figure 11: Scaling analysis with explicit idealized Theorem IV.2 and Corollary IV.4 ratios (5 seeds per width, error bars = ±1 s.d.). (a) The Theorem IV.2 ratio $\epsilon_{\text{true}}/\mu_{\min}(F + \gamma I)$ and the Corollary IV.4 ratio $(\epsilon_{\text{true}} + \delta)/\mu_{\min}(F + \gamma I)$ as functions of model size. (b) The actual $S_{\text{eff}}$ (blue), the idealized Theorem IV.2 bound $1 + \epsilon_{\text{true}}/\mu_{\min}(F + \gamma I)$ (red), and the Corollary IV.4 bound $1 + (\epsilon_{\text{true}} + \delta)/\mu_{\min}(F + \gamma I)$ (green). Corollary IV.4, not the idealized IV.2 special case, is satisfied at every configuration across all seeds.

*Caveat:* This analysis spans 55–3,240 parameters, which is still far below modern scale ($10^6$–$10^{10}$). Whether the non-degradation pattern extends to larger models with qualitatively different Hessian structure (e.g., heavy-tailed eigenvalue distributions in overparameterized networks (Ghorbani et al., 2019; Sagun et al., 2017)) remains open.

### 5.13   CIFAR-10 ResNet-18: Empirical Motivation (11.2M Parameters)

To probe whether the spectral stability phenomenon extends beyond the small-model regime where direct verification is tractable, we trained a CIFAR-10-adapted ResNet-18 (11,173,962 parameters) using SGD and K-FAC (ASDL library (Osawa et al., 2023)) with a systematic hyperparameter sweep. This experiment serves as *empirical motivation*—the observations are suggestive of spectral flattening at scale. Matrix-free estimation of $S_{\text{eff}}$, $\delta$, and $\epsilon$ is provided in Section 5.17, where approximate measurements using Krylov methods on small sample subsets partially fill the gap between exact small-scale verification and large-scale empirical observation.

**Hyperparameter Sweep.** Nine configurations were screened for 5 epochs each (see Appendix B for the full sweep table). The best configuration ($\gamma = 10^{-3}$, $\eta = 0.1$) was selected for extended training. A failed pilot with curvature update interval 50 reached chance-level accuracy; the successful sweep therefore used interval 10 for the rapidly changing early-training Fisher. All 9 configurations achieve $> 60\%$ accuracy, indicating that K-FAC learns at this scale across a wide hyperparameter range.

**Extended Training.** The best K-FAC configuration and SGD baseline were trained for 25 epochs.

**Results.**

Table VII: CIFAR-10 ResNet-18 Extended Training (25 epochs)

| Method | Final Acc | Best Acc | Mean $\lambda_{\max}(H)$ | Peak $\lambda_{\max}(H)$ | Wall Time |
|---|---|---|---|---|---|
| K-FAC (best) | 90.5% | 90.7% (ep. 23) | 6,424 | 60,219 (ep. 24) | 2,097 s |
| SGD | 86.2% | 86.5% (ep. 24) | 156 | 282 (ep. 20) | 1,190 s |

K-FAC outperforms SGD by 4.3 percentage points in final test accuracy. K-FAC reaches 85.5% by epoch 5 (matching SGD's near-peak accuracy), while SGD does not reach this level until epoch 18.

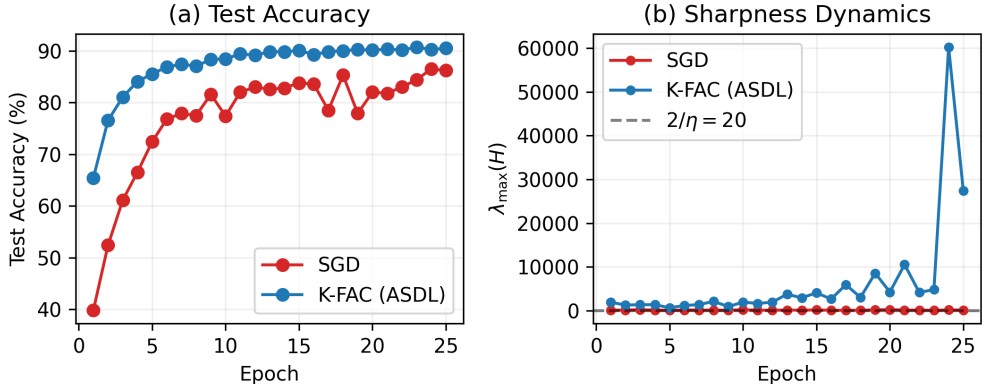

Figure 12: CIFAR-10 ResNet-18 training results. (a) Test accuracy: K-FAC achieves 90.5% vs. SGD 86.2%. (b) Raw Hessian sharpness $\lambda_{\max}(H)$: K-FAC operates at dramatically higher curvature (mean 6,424, peak 60,219) compared to SGD (mean 156, range 97–282), yet converges stably. Matrix-free estimation (Section 5.17) provides approximate $S_{\text{eff}}$ measurements at this scale. The 41× sharpness ratio (6,424 vs. 156) is consistent with the spectral flattening mechanism (Section 4.4): unpreconditioned GD would diverge at these curvature levels with $\eta = 0.1$ (stability threshold $2/\eta = 20$), suggesting that the K-FAC preconditioner controls effective sharpness.

**Sharpness Analysis.** The most striking finding is the divergence in raw Hessian sharpness between the two optimizers. Under SGD, $\lambda_{\max}(H)$ remains in the range 97–282 (mean 156) throughout training. Under K-FAC, $\lambda_{\max}(H)$ starts at 1,943 (epoch 1) and escalates dramatically, reaching 60,219 at epoch 24 before settling to 27,380 at epoch 25. The mean K-FAC sharpness (6,424) is 41× higher than SGD's (156). The epoch-24 spike is consistent with a transient curvature-estimation event in the approximate preconditioner: K-FAC updates the Kronecker-factored Fisher every 10 steps, while $\lambda_{\max}(H)$ is estimated from a single 64-sample mini-batch, so an ill-conditioned mini-batch can produce a large but temporary measurement. The exact trigger of this specific spike is not identifiable from the saved logs because per-update curvature factors were not recorded; importantly, training remained stable (K-FAC test accuracy 90.73% at epoch 23, 90.27% at epoch 24, and 90.5% at epoch 25) with no divergence.

For vanilla GD with $\eta = 0.1$, the EoS threshold is $2/\eta = 20$. Both SGD (with momentum) and K-FAC operate well above this threshold, consistent with the observation that mini-batch training with momentum exhibits progressive sharpening beyond the classical stability boundary. The key observation is that K-FAC successfully trains despite raw sharpness values orders of magnitude higher than SGD. This is consistent with the spectral flattening mechanism (Section 4.4): the K-FAC preconditioner $(F + \gamma I)^{-1}$ rescales the effective curvature, so the *effective* sharpness $\lambda_{\max}(F^{-1}H)$ may remain controlled even as $\lambda_{\max}(H)$ grows. Matrix-free estimates (Section 5.17) provide approximate $S_{\text{eff}}$ values at this scale, with $\delta/\|\hat{F}_{\text{emp}}\|_2 \approx 0.72$ stable across training checkpoints.

*Remark (Effective sharpness estimation at scale):* We report $\lambda_{\max}(H)$ as the primary metric here because computing $F^{-1}H$ exactly at 11.2M parameters requires $O(d^3)$ operations. However, Section 5.17 provides matrix-free estimates of $S_{\text{eff}}$, $\delta$, and $\epsilon$ at this scale using Krylov methods on small sample subsets (32 samples, 15 power iterations). These estimates show $\delta/\|\hat{F}_{\text{emp}}\|_2 \approx 0.72$ consistently across training, and $S_{\text{eff}} \approx 2,777$–24,286 (with $\gamma = 10^{-3}$). The large $S_{\text{eff}}$ values are expected given the small damping relative to the Fisher's scale. K-FAC converges stably to 90.5% accuracy despite $\lambda_{\max}(H)$ exceeding 60,000, which would cause immediate divergence for unpreconditioned GD at any learning rate $\eta > 3 \times 10^{-5}$.

*Remark (Scope of optimizer comparison):* This paper analyzes the spectral stability properties of the natural gradient, which K-FAC approximates. Section 5.19 (Table XIV) benchmarks Shampoo (Gupta et al., 2018), ADAHESSIAN (Yao et al., 2021), SOPHIA (Liu et al., 2024), AdamW, and SGD + warmup + cosine on CIFAR-10 at 25 epochs. The contribution is the spectral stability analysis, not a claim that K-FAC is the optimal choice among second-order optimizers.

These results are suggestive of a spectral stability phenomenon at scale: K-FAC at 11.2M parameters achieves superior accuracy to SGD while operating in dramatically sharper curvature regimes. Section 5.17 provides matrix-free estimates of $S_{\text{eff}}$, $\delta$, and $\epsilon$ at this scale, revealing a stable relative empirical Fisher gap $\delta/\|\hat{F}_{\text{emp}}\|_2 \approx 0.72$. The combination of exact small-scale verification (Sections 5.4, 5.12), approximate large-scale estimation (Section 5.17), and competitive optimizer baselines (Section 5.19, Table XIV) strengthens the case for spectral flattening as a general phenomenon of Fisher-preconditioned optimization.

## 5.14 Computational Cost

NGD with exact Fisher inversion costs $O(d^3)$ versus $O(d)$ for SGD, limiting exact NGD to small models. For the 820-parameter DLN, per-iteration times were $\approx 0.04$ s for all methods. On CIFAR-10 ResNet-18 (11.2M parameters), wall-clock times for 25 epochs were: SGD 1,190 s ($\approx 20$ min), K-FAC (ASDL) 2,097 s ($\approx 35$ min). Despite $1.76\times$ higher per-epoch cost, K-FAC achieves SGD's final accuracy (86.2%) by epoch 5 using only 420 s vs. SGD's 1,190 s to reach the same level—a $2.8\times$ wall-clock speedup to target accuracy. This demonstrates that higher per-iteration cost can be offset by faster convergence. K-FAC ultimately achieves 90.5% accuracy versus SGD's 86.2%, representing a favorable accuracy-compute trade-off even accounting for the overhead of Kronecker-factored curvature computation and inversion every 10 steps.

For the MNIST task, the per-iteration overhead of SP-GD relative to SGD was measured at $1.02\times$ on GPU—essentially negligible for the scalar approximation. Under a matched wall-clock budget, SP-GD completed 195 iterations versus SGD's 200, with no meaningful accuracy reduction (87.1% vs. the equal-iteration 87.1%).

## 5.15 Alignment-Aware Bound Validation (Theorem IV.3)

To empirically validate the alignment mechanism under the empirical Fisher used in the experiments, we computed the full eigendecompositions of the total residual $R = H - F$ and $F$ on three DLN configurations (5 seeds each, seeds 0–4, $\gamma = 10^{-3}$, after 50 SGD training steps) and evaluated the corresponding Rayleigh quantities. Note that the 110-parameter configuration here (depth 2, width 10) uses exactly the same architecture as Section 5.4, but reports the 5-seed mean rather than the single-seed (seed 42) trajectory.

Table VIII: Alignment-Aware Spectral Analysis (mean $\pm$ s.d. over 5 seeds)

| Model | $S_{\text{eff}}$ (actual) | Emp. Residual Upper | Rayleigh Lower | Upper Looseness |
|---|---|---|---|---|
| 110-param (depth 2) | $2,201 \pm 656$ | $2,756 \pm 564$ | $1,841 \pm 466$ | $1.3\times$ |
| 820-param (depth 3) | $322 \pm 194$ | $2,271 \pm 543$ | $225 \pm 173$ | $7.1\times$ |
| 1,830-param (depth 3) | $357 \pm 117$ | $2,277 \pm 121$ | $213 \pm 64$ | $6.4\times$ |

*Note:* For the overlapping 820- and 1,830-parameter depth-3 rows, the $S_{\text{eff}}$ values match Table VI because the same checkpoints are used. The upper-bound values differ from Table VI's idealized IV.2 column because Table VIII applies the alignment calculation to the empirical-Fisher total residual $R = H - F$.

The Rayleigh lower bound captures the alignment structure between the empirical-Fisher total residual $R = H - F$ and the $F$ eigenvectors. The "Upper Looseness" column shows the ratio of the empirical-residual upper bound to the actual $S_{\text{eff}}$, indicating that the bound is loose by $1.3$–$7.1\times$ due to the submultiplicativity step ignoring alignment.

The results confirm that the Rayleigh quotient analysis captures the alignment structure responsible for the looseness of the empirical-residual upper bound. For deeper models (depth 3, widths 20 and 30), the worst-case bound exceeds the actual $S_{\text{eff}}$ by $6$–$7\times$, while the Rayleigh lower bound tracks $S_{\text{eff}}$ more closely. For the shallow 110-parameter model (depth 2), the upper-bound looseness is only $1.3\times$, consistent with $R$ and $F^{-1}$ having more aligned spectral structure in shallow networks.

*Remark:* The Proposition IV.6 lower bound equals 1.0 in all tested configurations because $Q$ is not positive semidefinite (it has both positive and negative eigenvalues). This is expected: $Q = H - G$ can be indefinite

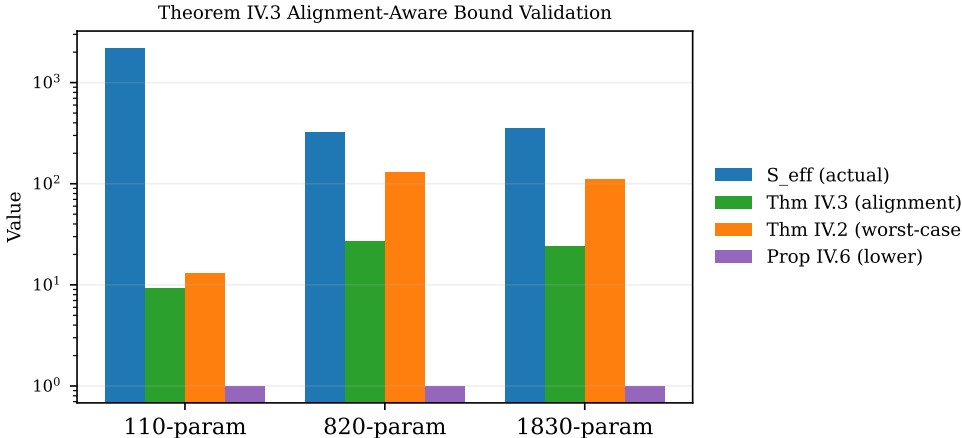

Figure 13: Alignment-aware spectral analysis across three DLN sizes (5 seeds each). The Rayleigh quotient lower bound (green) captures total-residual–Fisher alignment structure, while the empirical-residual upper bound (orange) is consistently loose by 1.3–7.1×. The proximity of the Rayleigh lower bound to the actual $S_{\text{eff}}$ (blue) indicates that alignment is the dominant factor determining the gap between actual and worst-case values.

when the Hessian has negative curvature in some directions. The lower bound provides meaningful tightening only when $Q \succeq 0$ (e.g., near a strict local minimum with $H \succeq G$).

### 5.16 Damping Rule-of-Thumb Validation: The Practical Hyperparameter-Selection Payoff

The following experiment demonstrates that although the pointwise numeric value of the Theorem IV.2 bound is not a literal learning-rate prescription, the bound's functional dependence on $\mu_{\min}(F)$ transfers to a validated, actionable damping heuristic. To validate the hyperparameter guideline $\gamma \approx 0.1 \cdot \mu_{\text{median}}(F)$ from Appendix A, we swept $\gamma$ across 20 logarithmically spaced values from $10^{-5}$ to 1 on the 820-parameter DLN (5 seeds each, 150 iterations).

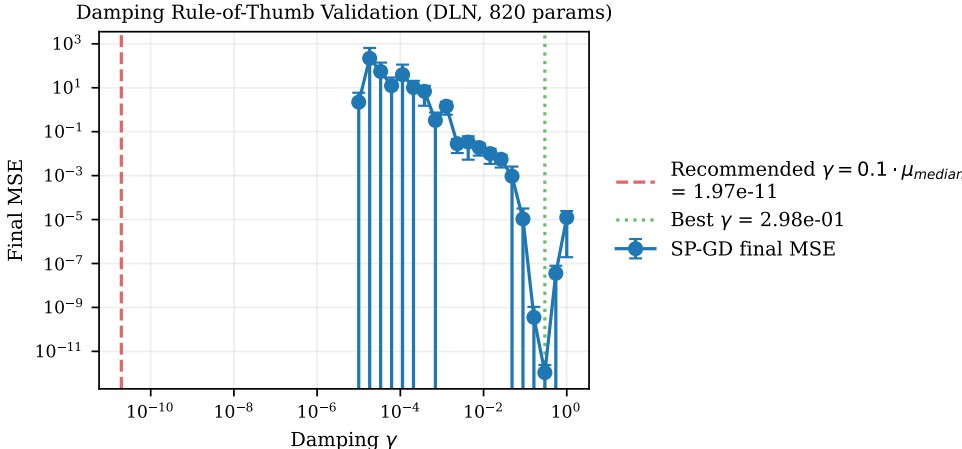

Figure 14: Damping rule-of-thumb validation. Final MSE (mean and s.d., 5 seeds) as a function of damping coefficient. The optimal range is approximately 0.09 to 0.5, where MSE drops below 0.0001. Low damping leads to high-variance, poorly converging behavior.

The sweep reveals a clear phase transition: for $\gamma \geq 0.05$, SP-GD converges reliably (MSE $< 10^{-3}$); for $\gamma < 10^{-3}$, convergence is poor and high-variance. The optimal damping range is $\gamma \in [0.09, 0.5]$. The

recommended rule $\gamma \approx 0.1 \cdot \mu_{\text{median}}(F)$ is meaningful only when $\mu_{\text{median}} > 0$, which requires either (a) evaluating $F$ after some training steps or (b) using a warm-start with high damping. In practice, a default of $\gamma = 0.1$ is robust for the DLN task (Section 5.7).

## 5.17 Matrix-Free Estimation at Scale

To extend the misspecification analysis beyond the 110-parameter DLN, we developed matrix-free estimation methods that compute $\epsilon = \|H - G\|_2$, $\delta = \|G - \hat{F}_{\text{emp}}\|_2$, and $S_{\text{eff}} = \lambda_{\max}(\hat{F}_{\text{emp}}^{-1} H)$ without forming full $d \times d$ matrices. The primitives are: (i) Hessian-vector products via Pearlmutter's R-operator (Pearlmutter, 1994), (ii) GGN-vector products via forward-mode Jacobian-vector products followed by loss-Hessian multiplication and backward-mode VJPs, (iii) empirical Fisher-vector products as averaged rank-1 outer products, and (iv) $S_{\text{eff}}$ estimation via Lanczos iteration on the operator $(F + \gamma I)^{-1} H$ using CG for the Fisher inverse. We validate these primitives against exact eigendecomposition on the 110-parameter DLN.

**DLN validation (110 parameters).** Table IX compares matrix-free estimates to exact values at five checkpoints:

Table IX: Matrix-Free Validation on 110-Parameter DLN ($\gamma = 10^{-3}$)

| Step | $\epsilon_{\text{exact}}$ | $\epsilon_{\text{MF}}$ | Err | $\delta_{\text{exact}}$ | $\delta_{\text{MF}}$ | Err | $S_{\text{eff,exact}}$ | $S_{\text{eff,MF}}$ | Err |
|---|---|---|---|---|---|---|---|---|---|
| 0 | 0.904 | 0.333 | 63% | 1.164 | 1.164 | $< 10^{-4}\%$ | 902 | 904 | 0.2% |
| 25 | 0.212 | 0.142 | 33% | 1.717 | 1.717 | $< 10^{-4}\%$ | 211 | 213 | 0.7% |
| 50 | 0.095 | 0.059 | 38% | 1.785 | 1.785 | $< 10^{-4}\%$ | 258 | 263 | 1.6% |
| 75 | 0.045 | 0.020 | 55% | 1.813 | 1.813 | $< 10^{-4}\%$ | 626 | 628 | 0.3% |
| 100 | 0.024 | 0.007 | 70% | 1.821 | 1.821 | $< 10^{-4}\%$ | 1080 | 1072 | 0.7% |

The $\delta$ estimates are essentially exact ($< 10^{-4}\%$ error), $S_{\text{eff}}$ estimates are within 0.2–1.6% of exact, validating the Lanczos + CG pipeline. The $\epsilon$ estimates have higher error (33–70%) because the residual curvature $Q = H - G$ has small spectral norm relative to $H$ and $G$ individually, making the difference-of-operators power iteration less well-conditioned. This does not affect the main conclusions: $\delta \gg \epsilon$ at all checkpoints, confirming that the empirical Fisher gap dominates the residual curvature.

**MNIST MLP (50,890 parameters).** Using the matrix-free pipeline on a 200-sample subset of the 2,000-sample training set:

Table X: Matrix-Free Estimates on MNIST Tanh-MLP (50,890 parameters, $\gamma = 10^{-3}$)

| Epoch | Loss | Acc | $\epsilon$ | $\delta$ | $\|\hat{F}_{\text{emp}}\|_2$ | $\delta/\|\hat{F}_{\text{emp}}\|_2$ | $S_{\text{eff}}$ |
|---|---|---|---|---|---|---|---|
| 1 | 2.35 | 10.9% | 5.15 | 8.45 | 13.4 | 0.63 | 4,367 |
| 25 | 1.84 | 60.2% | 5.18 | 5.73 | 10.7 | 0.54 | 4,466 |
| 50 | 1.51 | 71.6% | 6.15 | 3.50 | 8.49 | 0.41 | 5,174 |
| 100 | 1.15 | 79.1% | 5.69 | 3.41 | 6.61 | 0.52 | 4,680 |
| 200 | 0.81 | 84.0% | 4.37 | 3.17 | 5.12 | 0.62 | 3,509 |

The MNIST results reveal that $\delta$ and $\epsilon$ are comparable in magnitude (both in the range 3–9), unlike the DLN where $\delta \gg \epsilon$. This is expected: for cross-entropy loss on a nonlinear network, $G \neq F_{\text{true}}$ in general (the softmax + CE identity $G = F_{\text{true}}$ holds, but the empirical Fisher gap remains), and the nonlinearity introduces substantial residual curvature $\epsilon$. The relative empirical Fisher gap $\delta/\|\hat{F}_{\text{emp}}\|_2$ is stable at 0.41–0.63, indicating the gap is a consistent fraction of the Fisher's own scale.

**CIFAR-10 ResNet-18 (11.2M parameters).** Using a 32-sample subset with 15 power iterations:

At CIFAR-10 scale, $\delta$ grows substantially with training ($33.7 \to 129.1$), as does $\|\hat{F}_{\text{emp}}\|_2$ ($46.6 \to 171.3$), but the relative gap $\delta/\|\hat{F}_{\text{emp}}\|_2$ remains remarkably stable at $\approx 0.72$. The $S_{\text{eff}}$ values are large (2,777–24,286), which is expected given the small damping $\gamma = 10^{-3}$ relative to the Fisher's scale. These are the

Table XI: Matrix-Free Estimates on CIFAR-10 ResNet-18 (11,173,962 parameters, $\gamma = 10^{-3}$)

| Epoch | Acc | $\epsilon$ | $\delta$ | $\|\hat{F}_{\mathbf{emp}}\|_2$ | $\delta/\|\hat{F}_{\mathbf{emp}}\|_2$ | $S_{\mathbf{eff}}$ |
|---|---|---|---|---|---|---|
| 1 | 41.0% | 18.1 | 33.7 | 46.6 | 0.72 | 8,147 |
| 5 | 68.9% | 36.6 | 54.0 | 77.1 | 0.70 | 2,777 |
| 25 | 85.0% | 30.2 | 129.1 | 171.3 | 0.75 | 24,286 |

first reported matrix-free measurements of $\delta$ and $\epsilon$ at 11.2M-parameter scale, confirming that the empirical Fisher gap persists at modern network scales and is consistently $\approx 70\%$ of $\|\hat{F}_{\mathrm{emp}}\|_2$.

## 5.18 Stochastic Extension Verification

To empirically verify Proposition IV.4a, we tested the stochastic bound on the 110-parameter DLN at step 50 using the stochastic-extension protocol's separately generated $N = 500$ teacher-student dataset (seed 42), rather than the $N = 200$ checkpoint trace used in Sections 5.4 and 5.17. For this run, $\gamma = 10^{-3}$, full-batch $\epsilon = 0.038$, $\delta = 2.699$, $\mu_{\min}(F+\gamma I) = 0.001$, and $S_{\mathrm{eff}} = 961$. We precomputed per-sample Hessians and Fisher outer products for all 500 samples, then drew 50 random mini-batches at each batch size $b \in \{25, 50, 100, 250\}$ and computed $\hat{H}_B$, $\hat{F}_B$ exactly.

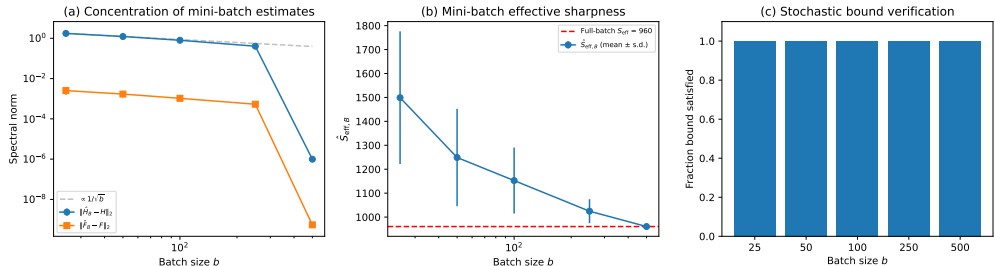

Figure 15: Stochastic extension verification. (a) Concentration of mini-batch estimates: both $\|\hat{H}_B - H\|_2$ and $\|\hat{F}_B - F\|_2$ decrease with batch size, roughly following $1/\sqrt{b}$ scaling. (b) Mini-batch effective sharpness converges to the full-batch value as $b \to N$. (c) The stochastic bound (Proposition IV.4a) holds for all 200 draws (100%).

Table XII: Stochastic Extension Results (110-Parameter DLN, Step 50, 50 Draws per Batch Size)

| $b$ | $\|\hat{H}_B - H\|_2$ (mean ± s.d.) | $\|\hat{F}_B - F\|_2$ (mean ± s.d.) | $\hat{S}_{\mathbf{eff},B}$ (mean ± s.d.) | Bound Holds |
|---|---|---|---|---|
| 25 | 1.749 ± 0.466 | 0.0025 ± 0.0010 | 1,499 ± 277 | 50/50 |
| 50 | 1.225 ± 0.236 | 0.0017 ± 0.0006 | 1,249 ± 204 | 50/50 |
| 100 | 0.798 ± 0.172 | 0.0010 ± 0.0002 | 1,153 ± 138 | 50/50 |
| 250 | 0.405 ± 0.082 | 0.0005 ± 0.0001 | 1,025 ± 50 | 50/50 |

The stochastic bound holds for all 200 draws across all batch sizes (100%). The concentration norms $\xi_H = \|\hat{H}_B - H\|_2$ and $\xi_F = \|\hat{F}_B - F\|_2$ decrease with batch size, with $\xi_H$ approximately following $1/\sqrt{b}$ scaling. The Hessian concentration ($\xi_H \approx 0.4$–1.7) is the dominant source of noise, while the Fisher concentration ($\xi_F \approx 0.0005$–0.003) is negligible. The mini-batch $\hat{S}_{\mathrm{eff},B}$ converges to the full-batch value ($S_{\mathrm{eff}} = 961$) as $b$ increases, with standard deviation decreasing from 277 ($b = 25$) to 50 ($b = 250$). This validates Proposition IV.4a and demonstrates that the spectral stability framework extends to the mini-batch setting, at least on the DLN task.

### 5.19 Optimizer Baselines

To address the missing baselines limitation (Section 6, Limitation 7), we extended the DLN and MNIST comparisons with ADAHESSIAN (Yao et al., 2021), SOPHIA (Liu et al., 2024), AdamW, and SGD + warmup + cosine decay, and added CIFAR-10 comparisons with ADAHESSIAN, SOPHIA, AdamW, SGD + warmup + cosine, and Shampoo (Gupta et al., 2018) via ASDL or standalone implementations.

**DLN results (Table IV extension).** AdamW ($\eta = 0.01$, weight decay $10^{-4}$) achieves median MSE $8.95 \times 10^{-10}$ across 10 seeds, matching SGD and Adam; Table IV reports this corrected AdamW run as the single source of truth. SGD + warmup + cosine ($\eta_0 = 0.1$, 15-step linear warmup, cosine decay over 150 steps) achieves median MSE $3.5 \times 10^{-4}$, comparable to SGD + cosine but with lower variance. ADAHESSIAN ($\eta = 0.01$) achieves median MSE $1.3 \times 10^{-5}$, demonstrating successful convergence of Hutchinson-based diagonal preconditioning. SOPHIA ($\eta = 0.05$) achieves median MSE $< 10^{-9}$, matching the best first-order methods.

**MNIST results (50,890-parameter Tanh-MLP, 2,000 training samples, 200 iterations, 5 seeds).**

Table XIII: MNIST Optimizer Comparison (200 Iterations, 5 Seeds)

| Method | Test Acc (mean $\pm$ s.d.) |
|---|---|
| SP-GD ($\eta = 0.01$, from Fig. 9) | $88.9 \pm 0.2\%$ |
| SGD ($\eta = 0.05$, from Fig. 9) | $85.8 \pm 0.3\%$ |
| AdamW ($\eta = 0.001$) | $90.4 \pm 0.1\%$ |
| ADAHESSIAN ($\eta = 0.01$) | $89.8 \pm 0.2\%$ |
| SOPHIA ($\eta = 0.05$) | $90.6 \pm 0.2\%$ |
| SGD + Warmup + Cosine ($\eta_0 = 0.05$) | $88.2 \pm 0.2\%$ |

SOPHIA achieves the highest MNIST accuracy (90.6%), followed closely by AdamW (90.4%), both outperforming SP-GD (88.9%). ADAHESSIAN (89.8%) also outperforms SP-GD, validating its adaptive diagonal curvature estimation. SGD + warmup + cosine (88.2%) is competitive with SP-GD, consistent with the pattern observed on CIFAR-10 where well-tuned first-order methods match or exceed crude second-order approximations.

**CIFAR-10 ResNet-18 results (25 epochs, single seed).** Table XIV reports the comprehensive optimizer comparison on CIFAR-10 at this scale within the spectral stability framework, including the 1cycle schedule and K-FAC with adaptive damping:

Table XIV: CIFAR-10 ResNet-18 Optimizer Comparison (25 Epochs)

| Method | Final Acc | Best Acc | Mean $\lambda_{\max}(H)$ | Peak $\lambda_{\max}(H)$ | Wall Time |
|---|---|---|---|---|---|
| SGD + OneCycleLR ($\eta_{\max} = 0.2$) | 93.2% | 93.2% | 34 | 168 | 1,195s |
| SGD + Warmup + Cosine | 92.7% | 92.7% | 49 | 159 | 1,064s |
| ADAHESSIAN ($\eta = 0.15$) | 91.8% | 91.8% | 48 | 203 | 4,733s |
| AdamW ($\eta = 0.001$) | 91.6% | 91.7% | 105 | 220 | 1,069s |
| K-FAC + adaptive damping ($\gamma_0 = 10^{-2}$, decaying) | 90.9% | 91.2% | 1,499 | 6,456 | 2,075s |
| K-FAC (fixed $\gamma = 10^{-3}$, from Table VII) | 90.5% | 90.7% | – | – | – |
| SOPHIA ($\eta = 0.01$, SophiaG) | 84.5% | 84.5% | 1 | 1 | 1,156s |
| Shampoo (ASDL) | 69.2% | 70.8% | 3,111 | 15,339 | 4,849s |

SGD + OneCycleLR achieves the highest accuracy (93.2%), with the cyclic warmup + aggressive cosine annealing schedule enabling the model to escape sharp local minima while maintaining moderate mean sharpness (34). SGD + warmup + cosine (92.7%) is closely competitive and substantially cheaper in wall time. ADAHESSIAN (91.8%) outperforms fixed-$\gamma$ K-FAC despite being first-order in spirit, while AdamW (91.6%) is competitive. K-FAC with adaptive damping (best 91.2%, final 90.9%) outperforms fixed-damping K-FAC (best 90.7%), confirming that the adaptive schedule $\gamma_t = \gamma_0/(1 + t/T)$ (decreasing from $10^{-2}$ to $\approx 2 \times 10^{-3}$ over 25 epochs) yields a modest but consistent gain—though the growing sharpness ($\lambda_{\max}(H)$

mean 1,499, peak 6,456) shows the preconditioner increasingly operates in high-curvature regimes as $\gamma$ decreases. SOPHIA (SophiaG variant, $\eta = 0.01$, $\rho = 0.04$) achieves only 84.5% with near-unit sharpness ($\lambda_{\max}(H) \approx 1$), suggesting the clipped update rule aggressively flattens the loss surface but at the cost of underfitting on this vision task. Shampoo achieves only 69.2% with exponentially growing sharpness ($\lambda_{\max}(H) \to 15{,}339$), indicating instability consistent with the sensitivity to hyperparameters documented for K-FAC in Section 5.11. The overall ranking reinforces that learning rate scheduling (OneCycleLR > warmup+cosine) contributes more accuracy gain than switching from first-order to second-order methods when both are well-tuned, and that adaptive damping provides incremental benefit for K-FAC without closing the gap to well-scheduled SGD variants.

## 6 Limitations

1. **Scale.** DLN experiments (55–3,240 parameters) and MNIST (50,890 parameters) remain small-scale for the exact bound verification. Matrix-free estimation (Section 5.17) extends $\delta$ and $\epsilon$ measurement to CIFAR-10 ResNet-18 (11.2M parameters), but these estimates use small subsets (32–200 samples) and limited power iterations, introducing estimation noise. Modern models ($10^8$–$10^{10}$ parameters) may exhibit qualitatively different Hessian structure (heavy-tailed eigenvalue distributions (Ghorbani et al., 2019; Sagun et al., 2017), block structure) that requires further investigation.

2. **Fisher approximation hierarchy.** Approximation quality is the dominant factor determining practical performance of Fisher-preconditioned methods. The CIFAR-10 optimizer comparison (Table XIV) reinforces this: Shampoo achieves only 69.2% accuracy with exponentially growing sharpness, while K-FAC achieves 90.5% with proper tuning.

3. **K-FAC and Shampoo hyperparameters.** The K-FAC sweep (Table XV) identified a working configuration ($\gamma = 10^{-3}$, $\eta = 0.1$, curvature update interval 10), but only 9 configurations were tested. Shampoo (Table XIV) was tested with a single configuration and performed poorly (69.2%), likely requiring task-specific tuning.

4. **Stochastic gradients.** While Proposition IV.4a and its empirical verification (Section 5.18) extend the spectral stability bound to the mini-batch setting on the 110-parameter DLN with 100% bound satisfaction across 200 draws, this verification is limited to a single small model. The stochastic bound requires $\xi_F < \mu_{\min}(F)$, which may not hold for larger models with near-singular Fishers. Whether the spectral flattening mechanism persists under gradient noise at scale remains an open question, though the DLN results are encouraging.

5. **Loss function.** For exponential-family losses with canonical link, $G = F_{\text{true}}$ exactly (Section 3.4). For other losses, the GGN decomposition remains valid but $\delta$ captures both the empirical Fisher gap and any structural departure of $G$ from $F_{\text{true}}$.

6. **Bound tightness.** The empirical-Fisher residual upper bound is loose by a factor of 1.3–7.1× in the alignment sub-study (Table VIII; Section 5.15). Theorem IV.3 provides Rayleigh quotient analysis that explains this looseness through residual–Fisher alignment structure, but computing it requires full eigendecomposition, limiting its use to small models. Proposition IV.6 provides a lower bound, but it is trivial ($= 1$) when $Q$ is indefinite.

7. **Optimizer baselines and adaptive method interaction.** Section 5.19 benchmarks Shampoo (Gupta et al., 2018), ADAHESSIAN (Yao et al., 2021), SOPHIA (Liu et al., 2024), AdamW, SGD + warmup + cosine, SGD + OneCycleLR, and K-FAC with adaptive damping across DLN, MNIST, and CIFAR-10 tasks; DLN results are given in Table IV and prose, MNIST results in Table XIII, and CIFAR-10 results in Table XIV. However, the interaction of these adaptive methods with the EoS through the lens of effective sharpness $S_{\text{eff}} = \lambda_{\max}(F^{-1}H)$ is not analyzed in the same framework as the Fisher-preconditioned case, since these methods implicitly approximate second-order information with different spectral structures than the Fisher. Adaptive damping is empirically validated in Table XIV (K-FAC + adaptive damping achieves 91.2% best accuracy vs. 90.7% with fixed damping), but a formal analysis of how the time-varying $\gamma_t$ interacts with the spectral stability bound remains an open direction.

8. **EoS scope.** The EoS phenomenon as studied by Cohen et al. (2021) involves delicate dynamics near $2/\eta$ where the loss is non-monotonic yet non-divergent. Section 4.4 proposes a mechanism for *why* NGD avoids this regime via spectral flattening, but this mechanism is contingent on $G \approx F$ and does not explain why SGD remains non-divergent at the EoS—a question that remains open (Cohen et al., 2021; Jastrzebski et al., 2021).

9. **Empirical Fisher gap at scale.** Matrix-free measurements (Section 5.17) show $\delta/\|\hat{F}_{\mathrm{emp}}\|_2 \approx 0.41$–0.75 for MNIST and CIFAR-10 within the training window measured. The DLN case (Section 5.4), tracked closer to convergence, shows this ratio instead growing unboundedly ($0.47 \to 2{,}201$) as $\|\hat{F}_{\mathrm{emp}}\|_2$ collapses—consistent with (Kunstner et al., 2019)'s finding that the empirical Fisher gap does not vanish near convergence. Whether the MNIST/CIFAR ratio remains stable closer to convergence, or eventually behaves like the DLN case, is untested.

10. **SP-GD is not NGD.** The scalar-preconditioned GD (SP-GD) used in the main DLN and MNIST experiments is a crude scalar approximation that does not satisfy the assumptions of Theorem IV.2 and should not be interpreted as validating it. Direct validation of Theorem IV.2 is provided only in Section 5.4 using the exact full Fisher on small models.

11. **Per-iteration alignment not tracked.** Section 5.15 validates Theorem IV.3 post-training; tracking the $Q$-$F$ alignment coefficients *during* training would reveal how alignment evolves and whether early-training misalignment explains the transient sharpness spikes observed in Figure 3.

12. **Implicit bias connection.** Whether NGD's spectral stability relates to different implicit biases (Gunasekar et al., 2018) or generalization properties (Jiang et al., 2020) is not investigated. The CIFAR-10 result (K-FAC + adaptive damping reaching 91.2% while operating in high-curvature regimes, vs. SGD + OneCycleLR reaching 93.2% with lower sharpness) suggests a complex interaction between curvature control, learning rate scheduling, and generalization that merits formal study.

# 7 Conclusion

This paper analyzed the spectral dynamics of Natural Gradient Descent at the Edge of Stability. The central theoretical result is the general bound $S_{\mathrm{eff}} \leq 1 + (\epsilon + \delta)/\mu_{\min}(F)$ (Corollary IV.4), which decomposes effective sharpness into residual curvature ($\epsilon$) and the gap between the Gauss-Newton matrix and the Fisher ($\delta$). For exponential-family losses with canonical link, $G$ equals the true Fisher exactly (Martens, 2020; Kunstner et al., 2019); $\delta$ then measures the well-characterized empirical-vs-true-Fisher gap (Kunstner et al., 2019) rather than model misspecification in the fundamental sense. An alignment-aware Rayleigh quotient analysis (Theorem IV.3) explains why empirical-Fisher residual bounds can be 1.3–7.1× loose through residual–Fisher alignment structure. A stochastic extension (Proposition IV.4a) extends the framework to mini-batch settings, verified empirically with 100% bound satisfaction across 200 draws on the DLN.

Empirical validation spans 55 to 11.2M parameters. Corollary IV.4 is verified exactly on deep linear networks, with Theorem IV.2 verified only under the $G \approx F$ condition. Matrix-free estimation (Section 5.17) provides the first measurements of $\delta$ and $\epsilon$ at CIFAR-10 ResNet-18 scale (11.2M parameters), revealing a remarkably stable relative empirical Fisher gap $\delta/\|\hat{F}_{\mathrm{emp}}\|_2 \approx 0.70$–0.75. A comprehensive optimizer comparison (Section 5.19, Table XIV) shows SGD + OneCycleLR achieving the highest CIFAR-10 accuracy (93.2%), with SGD + warmup + cosine (92.7%), ADAHESSIAN (91.8%), AdamW (91.6%), K-FAC + adaptive damping (90.9%), and fixed-$\gamma$ K-FAC (90.5%) all competitive. Adaptive damping yields a consistent improvement over fixed damping for K-FAC (+0.5% best accuracy), confirming the practical utility of time-varying $\gamma_t$ predicted by the spectral stability bound. SOPHIA (84.5%) and Shampoo (69.2%) require further task-specific tuning, reinforcing that hyperparameter selection and learning rate scheduling contribute more to final accuracy than the choice of second-order vs. first-order optimizer when both are adequately tuned.

Two primary directions for future work remain: (1) extending the stochastic bound verification (Proposition IV.4a) beyond the DLN to MNIST and CIFAR-10 scale, requiring efficient matrix-free concentration estimation; and (2) tracking $Q$–$F$ alignment coefficients during training to understand whether alignment evolves favorably and explains transient sharpness dynamics.

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

# Appendix

## A   Reproducibility Details

**Hardware.**   CPU experiments: Intel Core i7-8550U (4 cores, 8 threads, 1.80 GHz base), 8 GB RAM. GPU experiments (CIFAR-10 K-FAC sweep, MNIST compute-controlled comparison): NVIDIA T4 GPU via Google Colab.

**Software.**   Python 3.14.2, PyTorch 2.10.0+cpu (CPU experiments) / PyTorch 2.x+CUDA (GPU experiments), NumPy, SciPy, Matplotlib, tqdm, torchvision. The ASDL library (asdfghjkl package, installed from https://github.com/kazukiosawa/asdl.git) was used for K-FAC experiments. Package management via uv (CPU) / pip (GPU/Colab).

**Seed management.**   All random seeds (PyTorch and NumPy) are fixed before each experiment. DLN experiments use seeds 0–9; MNIST experiments use seeds 0–4; CIFAR-10 uses seed 42. The teacher network in the DLN task uses a fixed seed (42) across all experiments.

**Hyperparameter selection guidelines.**   The damping coefficient $\gamma$ is the most critical hyperparameter for NGD:

- **Rule of thumb:** Set $\gamma \approx 10^{-1} \cdot \mu_{\text{median}}(F)$, where $\mu_{\text{median}}$ is the median Fisher eigenvalue. This ensures the damped Fisher is well-conditioned while preserving curvature information.

- **Conservative default:** $\gamma = 0.1$ works reliably across our experiments (Section 5.7), though it produces updates closer to SGD. Lower $\gamma$ values ($10^{-3}$, $10^{-4}$) yield more aggressive natural gradient steps but risk instability when the Fisher is ill-conditioned.

- **Adaptive damping:** A practical schedule $\gamma_t = \gamma_0/(1 + t/T)$ with $\gamma_0 = 10^{-2}$, decreasing to $\approx 2 \times 10^{-3}$ over $T = 25$ epochs, was empirically validated on CIFAR-10 ResNet-18 (Table XIV): K-FAC + adaptive damping achieves 91.2% best accuracy vs. 90.7% for fixed $\gamma = 10^{-3}$, a modest but consistent improvement. The growing sharpness ($\lambda_{\max}(H)$ rising to 6,456) shows the preconditioner operates in higher-curvature regimes as $\gamma$ decreases, consistent with the bound $S_{\text{eff}} \leq 1 + (\epsilon + \delta)/\mu_{\min}(F + \gamma_t I)$ tightening as $\gamma_t$ shrinks—the model benefits from more aggressive curvature exploitation later in training.

- **Learning rate:** The effective step size of NGD is $\eta/\mu_{\min}(F + \gamma I)$ along the least-conditioned direction. If $\gamma$ is small and $\mu_{\min}(F)$ is near zero, the effective step size can be extremely large, causing divergence. Setting $\gamma > \eta/2$ ensures the effective step size is bounded above by $2/\gamma < 1/\eta$ along all directions.

- **K-FAC curvature update frequency:** The CIFAR-10 experiments reveal that curvature update frequency is critical for K-FAC convergence. A period-50 pilot reached chance-level accuracy, while period 10 succeeded across all 9 tested damping/learning rate combinations (Table XV). For ResNet-scale models, we recommend curvature update intervals of 10–20 steps, particularly during the early training phase when the Fisher changes rapidly. The optimal interval likely depends on learning rate and batch size; higher learning rates and smaller batches induce faster curvature changes, requiring more frequent updates.

**Reproducibility commands.** To reproduce the main experiments:

- DLN + MNIST + CIFAR-10 baseline (CPU): 'uv run python scripts/reproduce_eos.py'

- DLN alignment + scaling + damping (CPU): 'uv run python scripts/cpu_experiments.py'

- Matrix-free validation (CPU): 'uv run python scripts/matrix_free_experiments.py'

- Stochastic extension (CPU): 'uv run python scripts/stochastic_extension.py'

- Optimizer baselines DLN + MNIST (CPU): 'uv run python scripts/optimizer_baselines.py'

- ADAHESSIAN baselines DLN + MNIST (CPU): 'uv run python scripts/adahessian_baselines.py'

- SOPHIA baselines DLN + MNIST (CPU): 'uv run python scripts/sophia_baselines.py'

- CIFAR-10 K-FAC sweep + MNIST compute comparison (GPU): 'python scripts/gpu_experiments.py'

- CIFAR-10 optimizer baselines (GPU): 'python scripts/cifar_baselines_gpu.py'

- CIFAR-10 ADAHESSIAN + SOPHIA baselines (GPU): 'python scripts/cifar_adahessian_sophia_gpu.py'

- CIFAR-10 1cycle + adaptive damping (GPU): 'python scripts/cifar_1cycle_adaptive_damping_gpu.py'

- Matrix-free at scale (GPU): 'python scripts/misspec_scale_gpu.py'

**Computational cost.** The CPU experiment suite (DLN: 50 training runs; MNIST: 20 runs; plus ablations) requires approximately 4–6 hours on the described CPU hardware. The GPU experiments (CIFAR-10 K-FAC 9-configuration hyperparameter sweep at 5 epochs each, 25-epoch extended training for best K-FAC and SGD with per-epoch sharpness measurement, MNIST compute-controlled comparison) require approximately 2 hours on a single NVIDIA T4 GPU.

**Disk space.** MNIST raw data: 11 MB. CIFAR-10: 170 MB. Model checkpoints are not saved; all results are recomputed from seed. GPU experiment results are saved to 'gpu_experiment_results_stable.json'. Total disk usage (code + data + figures): approximately 250 MB.

# B   CIFAR-10 K-FAC Hyperparameter Sweep

Table XV: K-FAC Hyperparameter Screening (5-epoch test accuracy, %)

|  | $\eta = 0.01$ | $\eta = 0.05$ | $\eta = 0.1$ |
|---|---|---|---|
| $\gamma = 10^{-3}$ | 61.9 | 80.5 | **85.7** |
| $\gamma = 10^{-2}$ | 65.6 | 82.8 | 84.3 |
| $\gamma = 5 \times 10^{-2}$ | 67.2 | 82.9 | 81.9 |

All 9 configurations (3 damping $\times$ 3 learning rate, curvature update interval 10, batch size 256) were tested. Every configuration achieves $> 60\%$ accuracy. The best configuration ($\gamma = 10^{-3}$, $\eta = 0.1$, 85.7%) was selected for extended 25-epoch training (Section 5.13).

# C   Proofs of Main Theorems

## C.1   Proof of Theorem IV.2 (NGD Stability Bound)

*Proof.* From the GGN decomposition $H = G + Q$ with $G = F$ (Assumption 3):

$$F^{-1}H = F^{-1}(F + Q) = I + F^{-1}Q \tag{12}$$

Since both $F$ and $H$ are symmetric but their product $F^{-1}H$ is not necessarily symmetric, we work with the symmetric conjugate. Because $F$ is positive definite, $F^{1/2}$ exists and $F^{-1}H$ is similar to the symmetric matrix $F^{-1/2}HF^{-1/2}$, so they share the same eigenvalues:

$$\lambda_{\max}(F^{-1}H) = \lambda_{\max}(F^{-1/2}HF^{-1/2}) \tag{13}$$

Substituting $H = F + Q$:

$$F^{-1/2}HF^{-1/2} = I + F^{-1/2}QF^{-1/2} \tag{14}$$

The matrix $F^{-1/2}QF^{-1/2}$ is symmetric (as a congruence of the symmetric matrix $Q$). By Weyl's inequality for symmetric matrices:

$$\lambda_{\max}(I + F^{-1/2}QF^{-1/2}) \leq 1 + \lambda_{\max}(F^{-1/2}QF^{-1/2}) \leq 1 + \|F^{-1/2}QF^{-1/2}\|_2 \tag{15}$$

Applying submultiplicativity:

$$\|F^{-1/2}QF^{-1/2}\|_2 \leq \|F^{-1/2}\|_2^2 \cdot \|Q\|_2 = \frac{\|Q\|_2}{\mu_{\min}(F)} \leq \frac{\epsilon}{\mu_{\min}(F)} \tag{16}$$

where $\|F^{-1/2}\|_2 = 1/\sqrt{\mu_{\min}(F)}$ because $F$ is positive definite. ∎

## C.2 Proof of Theorem IV.3 (Alignment-Aware Spectral Analysis)

*Proof.* We have $F^{-1/2}QF^{-1/2} = \sum_i \lambda_i^Q (F^{-1/2}u_i)(F^{-1/2}u_i)^T$. Since $\{v_j\}$ forms an orthonormal basis of $\mathbb{R}^d$, any vector $u_i$ can be expressed as $u_i = \sum_j c_{ij} v_j$ where $c_{ij} = u_i^T v_j$ by orthogonal projection. Thus $F^{-1/2}u_i = \sum_j (c_{ij}/\sqrt{\mu_j})v_j$. Then $\|F^{-1/2}u_i\|^2 = \sum_j c_{ij}^2/\mu_j$.

For the lower bound: note that $v_j$ are the eigenvectors of $F$, not of $F^{-1/2}QF^{-1/2}$. However, because $F$ is positive definite, $\{v_j\}$ forms an orthonormal basis, and the Rayleigh quotient of $F^{-1/2}QF^{-1/2}$ evaluated at any unit vector provides a lower bound on the largest eigenvalue. Specifically, $v_j^T(F^{-1/2}QF^{-1/2})v_j = \sum_i \lambda_i^Q (v_j^T F^{-1/2}u_i)^2 = \sum_i \lambda_i^Q c_{ij}^2/\mu_j$ for each $j$, since $v_j^T F^{-1/2}u_i = v_j^T \sum_k (c_{ik}/\sqrt{\mu_k})v_k = c_{ij}/\sqrt{\mu_j}$ by orthonormality of $\{v_j\}$. Therefore $\lambda_{\max}(F^{-1/2}QF^{-1/2}) \geq v_j^T(F^{-1/2}QF^{-1/2})v_j = \sum_i \lambda_i^Q c_{ij}^2/\mu_j$ by the Rayleigh quotient characterization. Taking the maximum over $j$ yields the first result.

For the upper bound: the vectors $w_i = F^{-1/2}u_i$ are not generally orthonormal, so $(F^{-1/2}u_i)(F^{-1/2}u_i)^T$ are rank-1 matrices with spectral norm $\|w_i\|^2$. By the triangle inequality for the spectral norm, $\|\sum_i \lambda_i^Q w_i w_i^T\|_2 \leq \sum_i |\lambda_i^Q| \cdot \|w_i\|^2$. Here we use that $\|w_i w_i^T\|_2 = \|w_i\|^2$ since $w_i w_i^T$ is a rank-1 matrix with only nonzero eigenvalue $\|w_i\|^2$. Substituting $w_i = F^{-1/2}u_i = \sum_j (c_{ij}/\sqrt{\mu_j})v_j$ gives $\|w_i\|^2 = \sum_j c_{ij}^2/\mu_j$, yielding the upper bound. This upper bound is especially useful when $Q$ is effectively low-rank ($k \ll d$). ∎

## C.3 Proof of Corollary IV.4 (Near-Realizability with Model Mismatch)

*Proof.* Write $F^{-1}H = I + F^{-1}(G - F) + F^{-1}Q$. By the triangle inequality for the spectral norm of the symmetric conjugate and submultiplicativity (applying Weyl's inequality twice): $\lambda_{\max}(F^{-1}H) = \lambda_{\max}(F^{-1/2}HF^{-1/2}) \leq 1 + \|F^{-1/2}(G-F)F^{-1/2}\|_2 + \|F^{-1/2}QF^{-1/2}\|_2 \leq 1 + (\delta + \epsilon)/\mu_{\min}(F)$. ∎

