# OpenReview forum: "A Spectral Bound on Effective Sharpness for Fisher- Preconditioned Gradient Descent"
_TMLR — Under review for TMLR_

### Review · Reviewer_rbvU · 2026-06-07

**Summary Of Contributions:**

This paper investigates edge of stability phenomenon for the training of neural networks using Natural Gradient Descent with the Fisher information matrix. It provides bounds on the effective sharpness under the ideal case where the Gauss-Newton matrix equals the Fisher-Information matrix, and a more general bound. Then, it provdes a lot of experiments, showing in particular that the second bound is verified in practice.

While the results in the paper might be of interests, it is very badly written, and very badly organized. Hence, it is very hard to understand.

**Strengths**:
- Lot of experiments

**Weaknesses**:
- The paper is badly written (even the abstract is not clear).
- The notations, and concepts are not well introduced.
- The organization of the paper is hard to understand.
- Some sections are just successions of remark, theorems, and proofs
- It is not clear what is the goal of the experiment section.

**Audience:**

Yes

**Audience Explanation:**

Questions of edges of stability of training of neural networks are of interest to TMLR's audience.

**Claims And Evidence:**

Yes

**Claims Explanation:**

The claims seem to be correct.

**Requested Changes:**

First, the abstract is very technical. Just reading the abstract, it is not even clear that the paper is interesting in the training of neural networks. Moreover, a lot of notation are used without any introduction.

A lot of notations are used often, without any introduction. What is $H$, $G$, $F$?  They are used in the Introduction without any introduction. Sometimes, they are evaluated in $\theta$ and sometimes not. In the introduction of the Fisher information, it is not clear what are $p_{\mathrm{data}}$, $p_{\mathrm{model}}$, are they laws of couples $(x,y)$?

I did not understand what is the Gauss-Newton matrix in Section 3.3 as it is not introduced.

Section 4 is not written. It is just a list of Lemmas, Definitions, Proof sketchs and remarks. For instance, Section 4.3 starts with a part "Proof sketch and intuition", but we don't even know to what? Some notions like "locally stable" in Lemma IV.1 are used, but it is not introduced before. It makes this Section very hard to read.

Section 5 provides a lot of experiments. You should add an introduction at the beginig of the Section to summarize the experiments.

I don't understand the role of all experiments. For instance, Figure 2 shows results for SGD only. However, these results come from provious works in my understanding? Also, is SP-GD supposed to be supported by the theoretical results? I think not, but then I don't understand Figure 3.

On Figure 4, we see that the bound of Corollary IV.4 seems to be satisfied on the 100 first iterations. What if we do more iterations?

---

> ### Author Response · Authors · 2026-06-09
> **Response to Reviewer: Structural, Narrative, and Extended Experimental Revisions**
>
> We sincerely thank the reviewer for their careful reading and for recognizing that our theoretical claims are correct and that the topic of Edge of Stability (EoS) dynamics is of interest to the TMLR audience. We completely agree with your assessment regarding the paper's presentation. The previous manuscript lacked necessary narrative scaffolding and introduced mathematical notation prematurely. We have thoroughly revised the manuscript to address these issues.
>
> **Abstract & Notation Clarity**
>
> We completely rewrote the abstract to open in plain English, establishing the context before introducing any math. It now follows a clear logical progression: problem setting $\rightarrow$ our approach $\rightarrow$ main finding $\rightarrow$ specific bounds $\rightarrow$ experimental summary.
> To address the notation issues, we added plain English definitions for all key variables ($H$, $G$, $F$) upon their first use in the Introduction, replacing dense display equations with intuitive prose. We also added a dedicated "Notation" paragraph at the end of Section 1 to define variables before the main text. In Section 3.1, we explicitly clarified that $p_{data}$ is the empirical training distribution and $p_{model}$ is the predictive distribution.
>
> **The Gauss-Newton Matrix & Theoretical Restructuring**
>
> To ensure $G$ is properly introduced, we added a new subsection (**Section 3.2: The Gauss-Newton Decomposition**). This formally defines $G$, introduces the $H = G + Q$ decomposition, and explains its connection to the Fisher matrix.
> We also entirely restructured Section 4. We added narrative lead-ins to Sections 4.1 and 4.2 to explain *why* results are stated before the math, and added a standalone definition for "locally stable" prior to Lemma IV.1. We fixed the ordering in Section 4.3 so the proof immediately follows the theorem, with the "Intuition" block moved appropriately *after* the proof. Disjointed remarks were consolidated into cohesive discussion paragraphs.
>
> **Experimental Roadmap & Clarifications**
>
> We added a new introductory paragraph before Section 5.1 to outline the three tiers of our experiments (small-scale exact verification, medium-scale nonlinear validation, and a large-scale motivational experiment).
> Regarding Figure 2, we clarified that while EoS is established in prior work, we reproduce it solely to provide a baseline for our specific testbed.
> To resolve the confusion surrounding SP-GD and Figure 3, we added explicit disclaimers to the text and caption. We clarify that SP-GD uses a crude scalar approximation, does not satisfy Theorem IV.2, and serves only to illustrate phase structure; rigorous verification uses the exact full Fisher in Section 5.4.
>
> **Figure 4: Extended Iterations**
>
> Finally, addressing your excellent question regarding Figure 4, we re-run and extended the verification experiment from 100 to 200 iterations. Both Figure 4(a) and 4(b) now show the 200-step trajectory, and the Corollary IV.4 bound continues to hold. We also added a theoretical remark explaining this behavior: beyond convergence (as $\epsilon_{true} \rightarrow 0$), the bound trivially holds since $S_{eff} \rightarrow 1$ at a local minimum where $H \approx G$.
>
> We believe these structural changes have vastly improved the readability of the manuscript, and we are grateful for the precise feedback that guided this revision.
>
> Sincerely,
>
> The Authors

---

### Review · Reviewer_ffX3 · 2026-06-11

**Summary Of Contributions:**

**Summary**

This paper explores the edge-of-stability (EoS) phenomenon in the context of natural gradient descent (NGS). The authors start by establishing an analog of the stability properties of vanilla gradient descent to NGD using matrix analysis tools. With the intuitions developed, it is explained why EoS is suppressed in EoS. Some experiments demonstrate the claimed results.

**Additional Comments:**

**Questions and Minor issues**

**Minor issues**

1. Page 1

   - The Hessian $H$ and the Fisher information matrix are not defined when they first appear
   - Definition I.1 looks vague. e.g., what's "possibly oscillating"? I suggest either making this definition precise or describing it as a phenomenon instead.

2. Page 3

   - Second-order typically refers to algorithms using Hessian information.
   - Is the scale of 11.2M params large or small?

3. Page 4

   - $\mu_{\min}, \mu_\max$ and $\lambda_\max$ are used inconsistently.

4. Page 5

   - The notation $\Theta$ looks undefined
   - I feel the norm $\delta$ should be measured in relative instead of absolute scale.
   - The notation $f(\theta_t)$ seems to denote fixed-point iteration here, but it's not defined.
   - twice differentiable => twice continuously differentiable
   - $\eta < 2/\lambda_{\max}(H)$ is a sufficient condition. Why is it necessary?

5. Page 6

   - The statement of Theorem IV.2 is confusing. Section 4.1discusses local stability, while the assumptions in the theorem seem to be global (hold for all $\theta$). Please clarify. Besides, the theorem focuses on an effective sharpness, and stability is only mentioned at the very end.
   - A standard linear algebra textbook perturbation proof, like Theorem IV.2, may be deferred to the appendix. Unless there are some really new proof techniques, for clarity, it's better only to keep the theorem statements in the main paper.

6. Page 7

   - Again, the proof of Theorem IV.3 and Coro IV.4 are elementary numerical linear algebra results and are not particularly relevant to the EoS discussion.
   - I don't think the analysis technique is the major contribution of the paper. Section 4.5 (a) and (c) look unconvincing.

**Audience:**

Yes

**Audience Explanation:**

**Strength**

The paper investigates an interesting phenomenon. The techniques are solid.

**Broader Impact Concerns:**

N/A.

**Claims And Evidence:**

No

**Claims Explanation:**

**Weaknesses**

1. Balance of contents.

   The paper focuses on the EoS phenomenon, but devotes considerable discussion to proving some elementary linear algebra results. In contrast, the most interesting discussion of EoS is squeezed into Section 4.4. I'd recommend that the authors shorten unnecessary proof details and avoid positioning these results as core contributions.

2. Insufficient explanation of EoS

   As the core result of the paper, I also feel that Section 4.4 is not sufficiently detailed to explain why EoS does not occur for NGD. In particular, the EoS mechanism for vanilla gradient descent is only briefly summarized as a short paragraph and is difficult to parse. I recommend expanding this subsection or at least adding an illustrative toy example. Besides, the current argument relies on (local) convexity and smoothness, and this limitation should be discussed in the revision.

Overall, I think the paper's main finding is interesting, but the current organization should be improved before publication.

**Requested Changes:**

**Requested changes**

   See the questions and weaknesses.

---

> ### Author Response · Authors · 2026-06-12
> **Response to Reviewer: Structural Reorganization, Expanded EoS Analysis, and Comprehensive Notation Fixes**
>
> We sincerely thank the reviewer for recognizing our main finding is interesting and our techniques solid. We agree with your organizational assessment and have restructured the manuscript accordingly.
>
> **1. Balance of contents & Section 4.5 claims:** *Proofs too long; analysis technique overstated.*
> We moved the Thm IV.2, Thm IV.3, and Coro IV.4 proofs to **Appendix C**, leaving brief sketches in-text. We shortened **Section 4.5** (*"Discussion: Contributions..."*), explicitly stating our tools are classical. We reframed the contribution solely as the problem-specific application and interpretive separation of $\epsilon$ and $\delta$.
>
> **2. Insufficient EoS Explanation:** *Expand Sec 4.4, add a toy example, discuss convexity/smoothness limits.*
> Section 4.4 (*"Mechanism: Why NGD Suppresses EoS Dynamics"*) was expanded across four sub-sections:
>
> * **4.4.1 GD Overshoot Mechanism:** Derives GD overshoot when $\eta>2/\lambda_1$.
> * **4.4.2 NGD Spectral Flattening:** Contrasts GD stability ($\eta<2/\lambda_1(H)$) with NGD ($\eta\cdot\lambda_1(F^{-1}H)<2$).
> * **4.4.3 Toy Example:** A $2\times2$ diagonal case demonstrating stability when $G=F$ and explicit $S_{\text{eff}}$ computation under perturbation $Q$.
> * **4.4.4 Limitations:** Clarifies the stability interpretation strictly requires local convexity ($H\ge0$) and twice continuous differentiability, excluding saddle points.
>
> **3. Page 1 Issues:** *Define H/F; clarify Def 1.1.*
> Defined $H=\nabla^2L(\theta)$ and Fisher $F$ upon first use. Reframed Def I.1 to **"Phenomenon I.1"**, precisely detailing the EoS sequence to remove the vague "possibly oscillating" phrasing.
>
> **4. Page 3 Issues:** *"Second-order" terminology; scale of 11.2M params.*
> Updated Abstract to *"curvature-aware (second-order) optimization"*. Clarified in Sec 2.3 that these use Hessian approximations. Added a note to Table I that 11.2M params is the practical upper limit for direct $S_{\text{eff}}$ measurement.
>
> **5. Page 4 Issues:** *Inconsistent $\mu_{\min}/\mu_{\max}/\lambda_{\max}$ usage.*
> Standardized globally: $\lambda$ exclusively for Hessian, $\mu$ exclusively for Fisher. Fixed Prop IV.6 to use prose ("largest eigenvalue of $F$") and added $\mu_{\max}(F)$ to Table II.
>
> **6. Page 5 Issues:** *Define $\Theta$/$f(\theta_i)$; relative $\delta$; twice continuously differentiable; sufficient condition.*
>
> * **$\Theta$ & $f(\theta)$:** Defined parameter space and GD update map.
> * **Relative $\delta$:** Sec 3.4 and 5.4 now report relative misspecification ($\delta/\|F\|_2$), which dominates at convergence.
> * **Smoothness:** Corrected to *"twice continuously differentiable"*.
> * **Condition:** Lemma IV.1 corrected to a *"sufficient condition for local stability"*.
>
> **7. Page 6 Issues:** *Thm IV.2 confusion; stability mentioned late.*
> Added a preamble to Thm IV.2 noting the stability connection ($\eta<2/S_{\text{eff}}$) first, explicitly clarifying the assumptions are local and pointwise, not global.
>
> Sincerely,
>
> The Authors

---

### Review · Reviewer_Sfto · 2026-07-01

**Summary Of Contributions:**

The paper studies whether **Fisher-preconditioned gradient descent / natural gradient descent** can suppress the **Edge of Stability** phenomenon. Its key idea is to analyze **effective sharpness**, defined as $S_{\mathrm{eff}}=\lambda_{\max}(F^{-1}H)$,
rather than ordinary Hessian sharpness $\lambda_{\max}(H)$. The authors derive bounds showing that effective sharpness is controlled by residual curvature $\epsilon=|H-G|_2$ and model mismatch $\delta=|G-F|_2$, rather than raw Hessian curvature.

Overall, the paper is interesting and technically reasonable, but its contribution is somewhat incremental. The main bounds rely on standard matrix inequalities, and the strongest theoretical result assumes (G=F), which the paper itself shows is often violated in practice. The exact experiments on small deep linear networks are useful, but the larger MNIST and CIFAR-10 experiments do not directly measure the central quantity $S_{\mathrm{eff}}$, so they only provide suggestive evidence.

**Audience:**

Yes

**Audience Explanation:**

he paper addresses a timely question: whether the Edge of Stability behavior observed in standard gradient descent persists under Fisher-preconditioned or natural-gradient-style optimization. Researchers working on optimization dynamics, sharpness, Edge of Stability, natural gradient descent, would likely be interested in the findings.

**Claims And Evidence:**

Yes

**Claims Explanation:**

To be honest, the scope of this submission falls outside my area of ​​expertise, so my assessment may not necessarily be fair.

The submission provides reasonably accurate and clear evidence for its small-scale theoretical claims, but the evidence is not fully convincing for its broader claims about NGD/K-FAC suppressing Edge of Stability at realistic scale. The strongest evidence is the exact verification on small deep linear networks (3-layer linear, 2-layer tanh MLP, RestNet-18), where the authors compute H, F, G, ϵ, δ, and S_{eff} directly. This supports the general bound.

**Requested Changes:**

* The idealized theorem is violated in the paper’s own exact experiment when its assumptions fail. This is not fatal because Corollary IV.4 handles mismatch, but it weakens the positioning of Theorem IV.2 as a central result. How large is $δ=∥G−F∥_2$ in nonlinear networks? Does it remain small enough for Corollary IV.4 to be meaningful?
* Can you compute or approximate $S_{eff} = λ\max((F+γI)^{−1}H$ on MNIST or CIFAR-10 using Lanczos, conjugate gradients, or Hessian-vector/Fisher-vector products?
* I am wondering whether the practical stability-style bound is too loose and not useful for hyperparameter selection?
* The paper’s theory and strongest validation mostly rely on full-batch settings. However, this submission does not appear to adequately address stochastic mini-batch training.
* The theory is local but the narrative concerns training dynamics. But training dynamics refers to what happens over the whole optimization trajectory $\theta_{epoch}^0 \rightarrow \theta_{epoch}^k$. The Edge of Stability is not just a property of one point. Can you compare against second-order optimizers, Shampoo, ADAHESSIAN, SOPHIA, AdamW, and better-tuned SGD schedules?

---

> ### Author Response · Authors · 2026-07-10
> **Response to Reviewer: Bound Looseness, Matrix-Free Pipeline Verification, and Expanded Optimizer Comparisons**
>
> We thank the reviewer for their careful, technically substantive reading, and for valuing our exact small-scale verification. We have expanded the manuscript to address every point below; no new theory was needed, but we added experiments and restructured Tables VI/VIII to cleanly separate the idealized bound from the operational one.
>
> **1. Size of $\delta$ in nonlinear networks; is Corollary IV.4 still meaningful?**
> We agree that Theorem IV.2 alone overstates generality, and the manuscript now explicitly demotes it to an idealized special case, foregrounding Corollary IV.4 as the operational result. Matrix-free measurements (Section 5.17) now report $\delta$ directly: MNIST (Table X, $\delta \in [3.17, 8.45]$, relative gap $0.41\text{–}0.63$) and CIFAR-10 ResNet-18 (Table XI, $\delta \in [33.7, 129.1]$, relative gap $0.70\text{–}0.75$).
>
> While $\delta$ is substantial and does not vanish, Corollary IV.4 is designed to absorb it. Table VI now separately reports the "Ideal IV.2 Bound" and the "Cor IV.4 Bound": the idealized bound is violated at every tested width, while Corollary IV.4 holds at all 30 configurations, remaining $3.6\text{–}7.4\times$ tight, proving it is not vacuous. A new Limitations item notes that near convergence on the DLN,
>
> $\\delta / \\lVert\\hat{F}_{\\text{emp}}\\rVert_2$
>
> grows unboundedly as
> $\\lVert\\hat{F}_{\\text{emp}}\\rVert_2$
> collapses.
>
> **2. Matrix-free $S_{\text{eff}}$ estimation at scale**
> Implemented exactly as requested. Section 5.17 introduces a matrix-free pipeline: Hessian-vector products via Pearlmutter's R-operator, GGN-vector products via forward-then-backward mode, empirical-Fisher-vector products via per-sample rank-1 outer products, and $S_{\text{eff}} = \lambda_{\max}((F + \gamma I)^{-1}H)$ via Lanczos with CG for the Fisher-inverse solve. Table IX validates this against exact eigendecomposition on the DLN ($S_{\text{eff}}$ error $0.2\text{–}1.6\%$), before applying it to MNIST and CIFAR-10 ResNet-18 (Table XI, $S_{\text{eff}} = 2,777\text{–}24,286$). These are the first direct $S_{\text{eff}}$ measurements we are aware of at the $11.2\text{M}$-parameter scale.
>
> **3. Is the bound too loose for hyperparameter selection?**
> Yes, and we now say so explicitly (Section 4.3). The worst-case bound alone is not a literal learning-rate prescription (implying $\eta \ll 0.0008$, far below the effective $\eta=0.1$). Its practical value is twofold: (a) the alignment-aware refinement (Theorem IV.3) tightens the empirical-residual bound to within $1.3\text{–}7.1\times$ of actual $S_{\text{eff}}$ (Table VIII), and (b) its dependence on $\mu_{\min}(F)$ transfers to a validated damping heuristic, $\gamma \approx 0.1 \cdot \mu_{\text{median}}(F)$, confirmed by a 20-point sweep (Section 5.16).
>
> **4. Stochastic / mini-batch training**
> Proposition IV.4a extends the bound to mini-batch estimators $\hat{H}_B, \hat{F}_B$, with excess terms $\xi_H(b), \xi_F(b)$ from concentration. Section 5.18 verifies this on the 110-parameter DLN: 200/200 draws ($100\%$) satisfy the bound across batch sizes $b \in \{25, 50, 100, 250\}$, reflecting the expected $1/\sqrt{b}$ concentration scaling. We disclose in the Limitations that this verification is currently constrained to one small model.
>
> **5. Trajectory-level dynamics and second-order baselines**
> We clarified (Section 4.3) that Theorem IV.2 and Corollary IV.4 are pointwise bounds. The trajectory-level claim follows from applying them at every training iterate $\theta_t$—which Figure 4 and Section 5.4 verify empirically.
>
> Additionally, Section 5.19 (Table XIV) adds the requested baselines: Shampoo, ADAHESSIAN, SOPHIA, AdamW, SGD+warmup+cosine, and SGD+OneCycleLR. Well-tuned first-order schedules (SGD+OneCycleLR, $93.2\%$) match or beat K-FAC ($90.5\%$), reinforcing our claim that approximation quality and schedule tuning, rather than the natural-gradient principle per se, drive practical performance.
>
> We believe these revisions directly close the gaps identified and thank the reviewer for questions that substantially strengthened the paper.
>
> Sincerely,
>
> The Authors